# Differential activation of JAK-STAT signaling reveals functional compartmentalization in *Drosophila* blood progenitors

Diana Rodrigues[1,2,3], Yoan Renaud[4], K VijayRaghavan[2,3], Lucas Waltzer[4]*, Maneesha S Inamdar[1]*

[1]Jawaharlal Nehru Centre for Advanced Scientific Research, Bangalore, India; [2]National Centre for Biological Sciences, Tata Institute of Fundamental Research, Bangalore, India; [3]Shanmugha Arts, Science, Technology & Research Academy, Tamil Nadu, India; [4]University of Clermont Auvergne, CNRS, Inserm, GReD, Clermont-Ferrand, France

**Abstract** Blood cells arise from diverse pools of stem and progenitor cells. Understanding progenitor heterogeneity is a major challenge. The *Drosophila* larval lymph gland is a well-studied model to understand blood progenitor maintenance and recapitulates several aspects of vertebrate hematopoiesis. However in-depth analysis has focused on the anterior lobe progenitors (AP), ignoring the posterior progenitors (PP) from the posterior lobes. Using in situ expression mapping and developmental and transcriptome analysis, we reveal PP heterogeneity and identify molecular-genetic tools to study this abundant progenitor population. Functional analysis shows that PP resist differentiation upon immune challenge, in a JAK-STAT-dependent manner. Upon wasp parasitism, AP downregulate JAK-STAT signaling and form lamellocytes. In contrast, we show that PP activate STAT92E and remain undifferentiated, promoting survival. *Stat92E* knockdown or genetically reducing JAK-STAT signaling permits PP lamellocyte differentiation. We discuss how heterogeneity and compartmentalization allow functional segregation in response to systemic cues and could be widely applicable.

**\*For correspondence:**
lucas.waltzer@uca.fr (LW);
inamdar@jncasr.ac.in (MSI)

## Introduction

Blood and immune cells derive from hematopoietic stem cells (HSCs) that were classically thought to be a homogeneous population generating a defined hierarchy of progenitors (*Seita and Weissman, 2010*; *Yokota, 2019*). Recent studies reveal that mammalian HSCs and progenitor populations have dynamic cell surface marker phenotype and proliferative ability and varying in vivo differentiation potential in response to external cues (*Ema et al., 2014*; *Crisan and Dzierzak, 2016*; *Haas et al., 2018*). Studying aspects of intrapopulation heterogeneity and its implication in conditions of immune stress will improve our understanding of native and emergency hematopoiesis. Owing to the conserved signaling pathways and transcriptional factors that regulate hematopoiesis, *Drosophila melanogaster* has emerged as a powerful model to study blood cell development and the innate cellular immune response (*Banerjee et al., 2019*).

Like vertebrates, *Drosophila* hematopoiesis occurs in spatially and temporally distinct phases (*Tepass et al., 1994*; *Evans et al., 2003*; *Holz et al., 2003*; *Krzemien et al., 2010*). The embryonic wave of hematopoiesis primarily gives rise to macrophage-like phagocytic circulating and sessile hemocytes. A second wave of hematopoiesis takes place in a specialized larval hematopoietic organ, the lymph gland, located dorsally along the anterior cardiac tube (*Lanot et al., 2001*; *Mandal et al.,*

*2004*; *Grigorian and Hartenstein, 2013*; *Rugendorff et al., 1994*). In third instar larvae, the mature lymph gland is composed of a pair of anterior or primary lobes in segment T3/A1, followed by two to three pairs of lobes referred to as the secondary, tertiary, and quaternary lobes – collectively called the posterior lobes (*Banerjee et al., 2019*). Based on morphology and molecular marker analysis, primary lobes are compartmentalized into distinct zones. The posterior signaling center (PSC), a small group of cells at the posterior tip of the primary lobes, specifically expresses Antennapedia (Antp) and acts as a signaling niche. The medullary zone (MZ), close to the cardiac tube, consists of multi-potent progenitors and is identified by expression of *Drosophila* E-cadherin (DE-cad), the complement-like protein Tep4 and reporters for the JAK-STAT pathway receptor Domeless (Dome). The peripheral cortical zone (CZ) consists of differentiated blood cells that are mainly phagocytic plasmatocytes identified by Nimrod C1 (NimC1/P1) expression and a few crystal cells that express Lozenge (Lz) and the prophenoloxidases (ProPO). In addition, intermediate progenitors (IZ) reside in the region between the MZ and the CZ; they are identified by the expression of *dome* reporter and early differentiation markers like Hemolectin (Hml) or Peroxidasin (Pxn) but lack the expression of late markers like P1 for plasmatocytes and Lz for crystal cells (*Banerjee et al., 2019*; *Jung et al., 2005*).

The presence of blood cell progenitors in the anterior lobes prompted intense investigations to unravel how their fate is controlled. During normal development, anterior progenitor (AP) proliferation, quiescence, and differentiation are finely orchestrated by the interplay of various pathways. Notably AP maintenance is controlled by activation of the Hedgehog and ADGF-A pathways in the MZ in response to signaling from the PSC and the CZ respectively (*Mandal et al., 2007*; *Baldeosingh et al., 2018*; *Mondal et al., 2011*). Moreover, within the MZ, reactive oxygen species (ROS) levels, Wingless signaling, and Collier expression regulate AP differentiation (*Owusu-Ansah and Banerjee, 2009*; *Sinenko et al., 2009*; *Benmimoun et al., 2015*). Besides, AP fate is controlled by systemic cues and external stresses such as immune challenge (*Krzemien et al., 2010*; *Khadilkar et al., 2017b*). In particular, deposition of egg from the parasitoid wasp *Leptopilina boulardi* (*L. boulardi*) in the hemocoel triggers AP differentiation into lamellocytes and premature histolysis of the primary lobes (*Lanot et al., 2001*; *Crozatier et al., 2004*; *Louradour et al., 2017*; *Benmimoun et al., 2015*; *Bazzi et al., 2018*; *Letourneau et al., 2016*; *Small et al., 2014*). ROS-mediated activation of EGFR and Toll/NF-kB signaling pathways in the PSC and the consecutive downregulation of the JAK-STAT pathway in the AP are essential for this cellular immune response (*Makki et al., 2010*; *Louradour et al., 2017*).

In contrast with the anterior lobes, little is known about the posterior progenitors (PP) present in the posterior lobes (*Banerjee et al., 2019*). The general view is that these lobes essentially harbor progenitors as initially suggested by the higher expression of DE-cadherin and the lack of expression of mature blood cell markers (*Jung et al., 2005*). Yet only few studies on progenitor differentiation in the primary lobe report on phenotypes in the secondary lobes (*Owusu-Ansah and Banerjee, 2009*; *Dragojlovic-Munther and Martinez-Agosto, 2013*; *Khadilkar et al., 2017b*; *Hao and Jin, 2017*; *Zhang and Cadigan, 2017*; *Kulkarni et al., 2011*; *Benmimoun et al., 2015*). These studies revealed that posterior lobes could also differentiate in genetic contexts where there is extensive premature differentiation in primary lobes; however, a thorough analysis of the posterior lymph gland lobes is lacking. In addition, depletion of *asrij*, *arf1*, *or garz* and overexpression of *arf1GAP*, all show more severe phenotypes of hyperproliferation and premature differentiation in the posterior lobes compared to the primary lobes (*Kulkarni et al., 2011*; *Khadilkar et al., 2014*). On the other hand, *Stat92E* mutation causes premature progenitor differentiation in the primary lobes but not in the posterior lobes (*Krzemień et al., 2007*), suggesting important differences in regulation and function of these lobes as well as inherent differences within the progenitor pool. Along that line, a recent single-cell RNA sequencing analysis of the lymph gland has identified novel hemocyte subpopulations and suggests a higher degree of blood cell progenitor heterogeneity than previously thought (*Cho et al., 2020*). However, the posterior lobes were excluded from this study and positional information is not retained in such single-cell sequencing analysis. A systematic mapping of lymph gland progenitors in vivo is not available, and we thus lack detailed lobe-wise information about these progenitors. The small size and contiguous nature of the *Drosophila* hematopoietic organ allows simultaneous comprehensive assessment of progenitors across lobes, which is essential for understanding the dynamics of progenitor heterogeneity and function in response to local and systemic cues. Such analysis is currently not feasible in vertebrate hematopoiesis. Here we combined in situ expression analysis of the entire lymph gland with differential RNA sequencing analysis of the

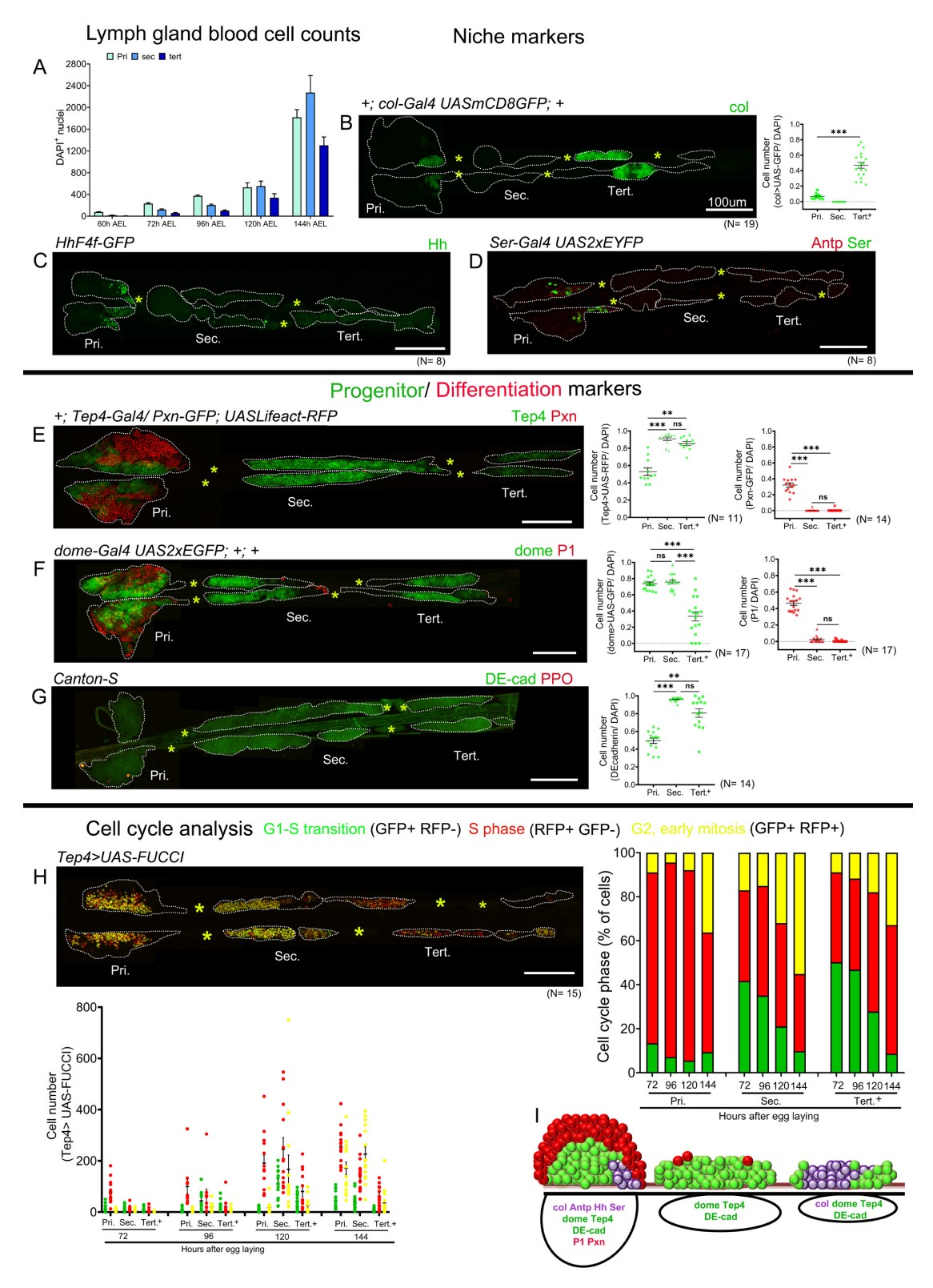

**Figure 1.** Blood cell counts and gene expression analysis in the larval lymph gland. Whole lymph gland preparations (including primary and posterior lobes [sec., tert., quat. lobes]) were analyzed for third instar larvae except where time points are mentioned. (**A**) Total lymph gland blood cell counts (DAPI+ cells) at the indicated time points. (**B–D**) Expression profiles of posterior signaling center (PSC) markers. (**B**) c*ol-Gal4,UASmCD8-GFP* (green) is expressed in the PSC and in the tertiary lobes; graph represents ratio of col>GFP+/DAPI+ cells. (**C**) *hhF4f-GFP* (green) and (**D**) Antp (red) and *Ser-Gal4,*

*Figure 1 continued on next page*

*Figure 1 continued*

*UAS2xEYFP* (green pseudo color) are expressed in the PSC only. (**E–G**) Expression profiles of medullary zone (MZ)/progenitor (green) and cortical zone (CZ)/differentiation (red) markers. (**E**) *Tep4-Gal4,UASLifeact-RFP* (green pseudo color), (**F**) *dome-Gal4,UAS2xEGFP* (green), and (**G**) DE-cadherin (green) expression are observed in the MZ as well as in the posterior lobes; (**E**) *Pxn-GFP* (red pseudo color), (**F**) P1 (red), and (**G**) ProPO (red) are restricted to the primary lobes. Quantification panels indicate ratios of Tep4$^+$/DAPI$^+$ or Pxn$^+$/DAPI$^+$ (**E**), dome$^+$/DAPI$^+$ or P1$^+$/DAPI$^+$ (**F**), and DE-cad$^+$/DAPI$^+$ (**G**) cells respectively. (**H**) Cell cycle analysis for *Tep4-Gal4> FUCCI*. GFP+/RFP-: G1 phase; GFP-/RFP+: S-phase, GFP+/RFP+: G2/M phase. (**I**) Schematic representation of the expression of indicated markers in the third instar larval lymph gland. Pri. indicates primary lobes, Sec. indicates secondary lobes, Tert. indicates tertiary lobes, and Post. indicates posterior lobes. (**B–D and E–H**) Yellow asterisks indicate pericardial cells. Lobes are outlined by white dashed lines. Nuclei were stained with DAPI, which is not displayed for clarity. (**B–G**) Kruskal–Wallis test was used for statistical analysis. **p<0.01, ***p<0.001, ns: nonsignificant, and error bars represent SEM. (**B–H**) Scale bar: 100 μm.

The online version of this article includes the following source data and figure supplement(s) for figure 1:

**Source data 1.** Numerical data plotted and statistical analysis related to *Figure 1*.
**Figure supplement 1.** Analysis of lymph gland lobes and gene expression analysis.
**Figure supplement 1—source data 1.** Numerical data plotted and statistical analysis related to *Figure 1—figure supplement 1*.
**Figure supplement 2.** Blood cell progenitor cell cycle pattern.
**Figure supplement 2—source data 1.** Numerical data plotted and statistical analysis related to *Figure 1—figure supplement 2*.

anterior and the posterior lobes to map expression of known blood cell markers and to identify new progenitor markers. Furthermore, by assessing the response to immune challenge, we reveal the functional heterogeneity of the lymph gland progenitors and we propose that differential regulation of the JAK-STAT pathway underlies the maintenance of PP in response to wasp infestation.

## Results

### Characterization of lymph gland anterior lobes and posterior lobes development and gene expression pattern

In third instar larvae, the lymph gland posterior lobes are separated from the primary lobes and from each other by pericardial cells. To shed further light on these poorly characterized lobes, we used a dissection method that preserves the whole lymph gland (see Materials and methods) and we assessed their formation at different times of larval development, from 60 hr after egg laying (AEL) (mid second instar) to 144 hr AEL (late wandering third instar). While one pair of secondary lobes is generally clearly visible from 60 hr AEL onward, we often distinguished only one tertiary lobe until 120 hr AEL, and quaternary lobes were rarely observed (*Figure 1—figure supplement 1A*). At 60 hr AEL, primary lobes contain around 80 cells, whereas the secondary lobes consist of approximately 10–15 cells and tertiary lobes are barely visible. As shown in *Figure 1A*, the number of cells in these lobes dramatically increases from 60 to 144 hr AEL. While the lymph gland secondary and tertiary lobes all together contain significantly less cells than the primary lobes from 60 to 96 hr, this ratio is then inverted so that posterior lobes together consist of about twice the number of cells present in the primary lobes at the end of larval life (*Figure 1A*, *Figure 1—figure supplement 1B–F*), indicating that they significantly contribute to the larval hematopoietic system.

To better define the identity of the blood cells in the posterior lobes, we then analyzed the expression of well-characterized PSC, MZ, and CZ markers. Consistent with previous reports, we observed that the *hedgehog* reporter line *hhF4f-GFP* (*Tokusumi et al., 2010*), the Gal4 lines for *Serrate* (*Ser-Gal4, UAS2xEYFP*) (*Lebestky et al., 2003*), and *collier* (*col-Gal4,UASmCD8-GFP*) (*Crozatier et al., 2004*) as well as the transcription factor Antennapedia (Antp) (*Mandal et al., 2007*) were expressed in the PSC. *col-Gal4-UASmCD8-GFP* is highly expressed in approximately 40% of the cells in the tertiary lobe (*Figure 1B*). However, *hhF4f-GFP*, *Ser-Gal4, UAS2xEYFP*, and Antp did not show any expression in posterior lobes (*Figure 1C,D*). Immunostaining also revealed high levels of Col in the PSC and tertiary lobes, while lower levels are detected in the MZ and the secondary lobes as reported earlier (*Figure 1—figure supplement 1G*; *Benmimoun et al., 2015*). To assess the distribution of progenitors, we analyzed MZ markers expression using Gal4 enhancer trap lines in *Thioester-containing protein 4* (*Tep4-Gal4,UASLifeact-RFP*) (*Avet-Rochex et al., 2010*) and *domeless* (*dome-Gal4,UAS2xEGFP*) (*Jung et al., 2005*) and immunostaining against *Drosophila* E-cadherin (DE-cadherin) (*Jung et al., 2005*). Besides their expression in the MZ of the primary

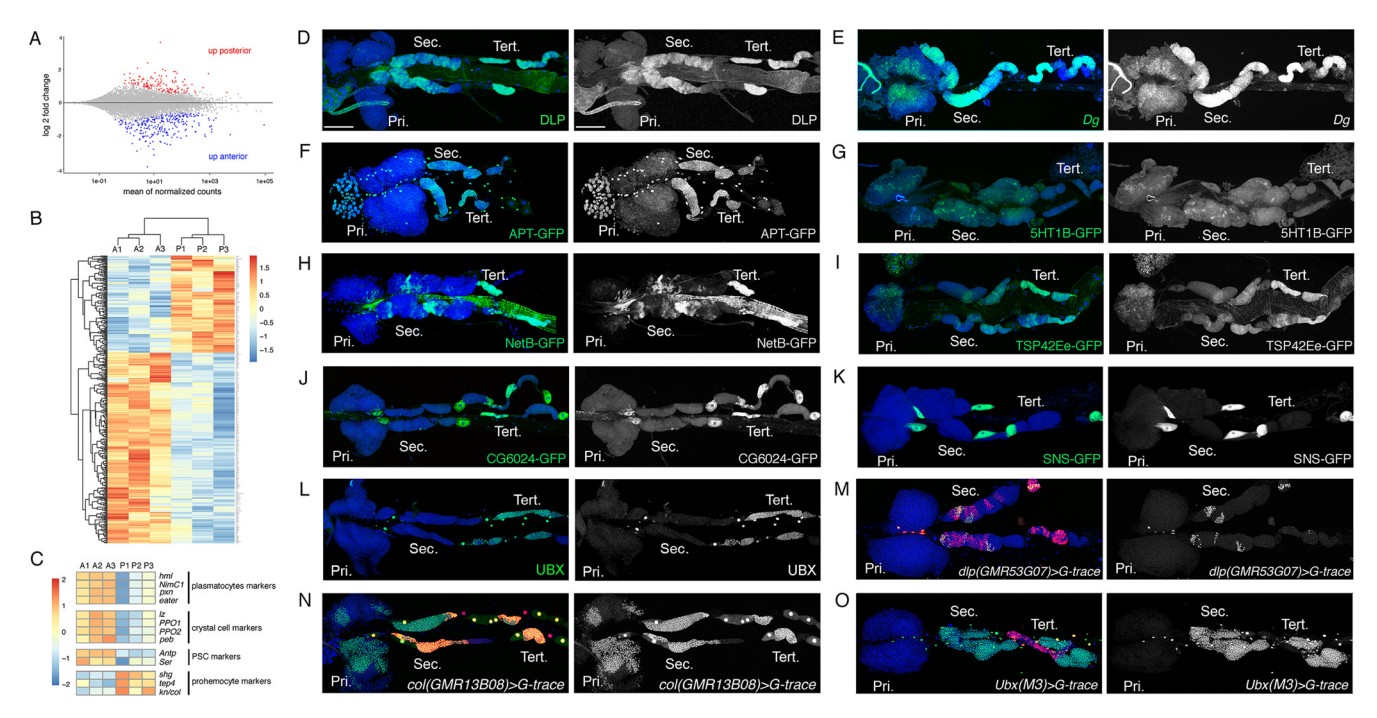

**Figure 2.** Characterization of the expression profile of the lymph gland anterior and posterior lobes and identification of new markers. (A) MA-plot of DESeq2 results for RNA-seq data comparison between anterior and posterior lymph gland lobes dissected from wandering third instar larvae. Genes that are differentially expressed (adjusted p<0.01 and fold-change >1.5) are highlighted in red (up in posterior lobes) or blue (up in anterior lobes). (B) Heat map of differentially expressed genes between the anterior (A1, A2, and A3) and posterior (P1, P2, and P3) lobes RNA-seq samples. Hierarchical clustering was performed using R-Bioconductor. (C) Heat map of gene expression between anterior and posterior lobes for selected markers of lymph gland blood cell populations. (D–O) Third instar larvae whole lymph gland preparations showing the expression of the indicated genes or knock-in GFP fusions as revealed by immunostaining or RNA in situ (D–L) or the G-traced (green) and live (red) expression of the indicated Gal4 drivers (M–O). Nuclei were stained with DAPI (blue). The right-hand panels display the green channel only. Pri. indicates primary lobes, Sec. indicates secondary lobes, and Tert. indicates tertiary lobes. Scale bar: 100 µm.

The online version of this article includes the following figure supplement(s) for figure 2:

**Figure supplement 1.** Expression of new markers in the lymph gland.

lobes, *Tep4-Gal4, dome-Gal4*, and DE-cadherin are expressed in most cells of the secondary lobes and some cell clusters in the tertiary and quaternary lobes (*Figure 1E–G*). Similar patterns were observed using RNA in situ hybridization against *Tep4* or GFP immunostaining against a Shg-GFP (DE-cadherin) endogenous fusion or the *domeMESO-GFP* reporter (*Figure 1—figure supplement 1H–J*). Differentiated cells reside primarily in the CZ and can be visualized by the expression of *Peroxidasin* (*Pxn-GFP*), an early marker of differentiation, NimC1 (P1) a plasmatocyte marker, or Prophenoloxidase (ProPO) and Hindsight/Pebbled (Hnt/Peb), two crystal cell markers. The analysis of the expression of these markers showed that differentiated cells are rare in the posterior lobes (*Figure 1E–G* and *Figure 1—figure supplement 1K*). Thus, our analysis provides conclusive evidence that the posterior lobes are made up almost entirely of undifferentiated progenitors and actually represent the main reservoir of lymph gland progenitors in the third instar larvae.

To gain further insight into the behavior of the AP and PPs, we also monitored their cell cycle parameters using the Fly FUCCI system (*Zielke et al., 2014*). Analysis of *Tep4*-expressing progenitors, which form the major fraction of the progenitor pool, showed that most of the AP are in S phases from 72 hr to 120 hr AEL and that their proliferation is reduced at 144 hr AEL (*Figure 1H* and *Figure 1—figure supplement 2A–C*). In contrast, fewer cells are in S phase among the PP, and the proportion of PP in G1 phase constantly decreases from 72 hr to 144 hr AEL essentially to the benefit of cells in G2/M, both in the secondary and tertiary lobes. Cell cycle of the entire pool of

lymph gland cells using *e33c-Gal4* driver showed that the overall distribution of proliferative cells was more in the tertiary lobes at 120 hr AEL (*Figure 1—figure supplement 2D–G*).

In sum, our data demonstrate that the posterior lobes represent a major pool of progenitors and reveal heterogeneity of gene expression and cell cycle pattern across lymph gland progenitor populations (*Figure 1I*).

## Anterior and posterior lobes of the larval lymph gland differ in their gene expression profile

To gain further insights into the anteroposterior compartmentalization of the larval lymph gland and to identify new blood cell markers, we established the transcriptome of the anterior and posterior lobes in wild-type third instar wandering larvae (see Materials and methods for details). Accordingly, the gene expression profile of manually-dissected anterior or posterior lobes was determined by RNA sequencing (RNA-seq) from biological triplicates using Illumina NextSeq550 sequencing system. We observed that 6709 genes (corresponding to ±38% of the genes on *Drosophila* reference genome dm6) are expressed with a RPKM >1 in all three samples of the anterior lobes or of the posterior lobes (*Supplementary file 1*), including well-known pan-hematopoietic markers such as *asrij* or *serpent* (*srp*). Using DESeq2, we found that 406 genes are differentially expressed (p<0.01 and fold change >1.5) between the anterior and the posterior lobes, with 269 genes overexpressed in the anterior lobes and 137 genes overexpressed in the posterior lobes (*Figure 2A,B*, *Supplementary file 2*). In line with previous studies and the above results, markers of differentiated blood cells such as *Hemolectin* (*Hml*), *NimC1, Pxn,* and *eater* for the plasmatocyte lineage, or *lz, PPO1, PPO2,* and *peb,* for the crystal cell lineage, as well as PSC markers such as *Antp* and *Ser,* were overexpressed in the anterior lobes, whereas blood cell progenitor markers such as *DE-cadherin (shg)*, *Tep4* or *col (col/kn)*, were overexpressed in the posterior lobes (*Figure 2C*). Gene ontology (GO) enrichment analyses showed a very strong over-representation for genes implicated in immune processes/defense responses and extracellular matrix (ECM) organization among the genes

**Table 1.** Main terms enriched in gene ontology (GO) analysis.

| | Up in anterior lobes | | Up in posterior lobes | |
|---|---|---|---|---|
| | p-value | #genes | p-value | # genes |
| **GO-term biological processes** | | | | |
| Response to stimulus | 6,88E-10 | 92 | 1,24E-05 | 47 |
| Extracellular matrix (ECM) organization | 1,97E-09 | 11 | ns | 0 |
| Immune system process | 1,09E-07 | 26 | ns | 4 |
| Cell–cell adhesion | ns | 6 | 9,75E-10 | 12 |
| Nephrocyte filtration | ns | 0 | 1,88E-07 | 0 |
| Neurogenesis | 1,30E-04 | 34 | 8,88E-06 | 20 |
| **GO-term cellular components** | | | | |
| Collagen-containing ECM | 2,21E-13 | 10 | ns | 1 |
| Cell–cell junction | ns | 5 | 3,28E-06 | 8 |
| Nephrocyte diaphragm | ns | 0 | 2,15E-06 | 4 |

overexpressed in the anterior lobes (*Table 1* and *Supplementary file 3*), an observation consistent with the roles of differentiated blood cells in immune response and ECM synthesis (*Banerjee et al., 2019*). In contrast, GO analysis on the genes overexpressed in the posterior lobes were biased toward cell adhesion, neuronal differentiation, and nephrocyte filtration (*Table 1* and *Supplementary file 3*). While the latter GO enrichment is likely due to the presence of pericardial cells in dissected posterior lobes (see below), the neuronal link is unexpected and certainly warrants future investigations.

To validate our transcriptomic data and identify new markers for blood cell progenitors and/or the posterior lobes, we then analyzed the expression of several genes that were overexpressed in the posterior lobes according to our RNA-seq. Immunostaining against the heparan sulfate proteoglycan (HSPG) Dally-like protein (Dlp), which was described as a PSC marker (*Pennetier et al., 2012*) (see also *Figure 2—figure supplement 1A*), revealed that it is expressed at high levels in the posterior lobes and at very low levels in the MZ of the anterior lobes (*Figure 2D*). Consistent with the idea that its expression is activated by Col (*Hao and Jin, 2017*), Dlp displayed an anteroposterior gradient of expression similar to that of *col* (*Figure 2D*), with particularly strong levels in the tertiary lobes. RNA in situ hybridization against *Dystroglycan* (*Dg*), which encodes a cell surface receptor for

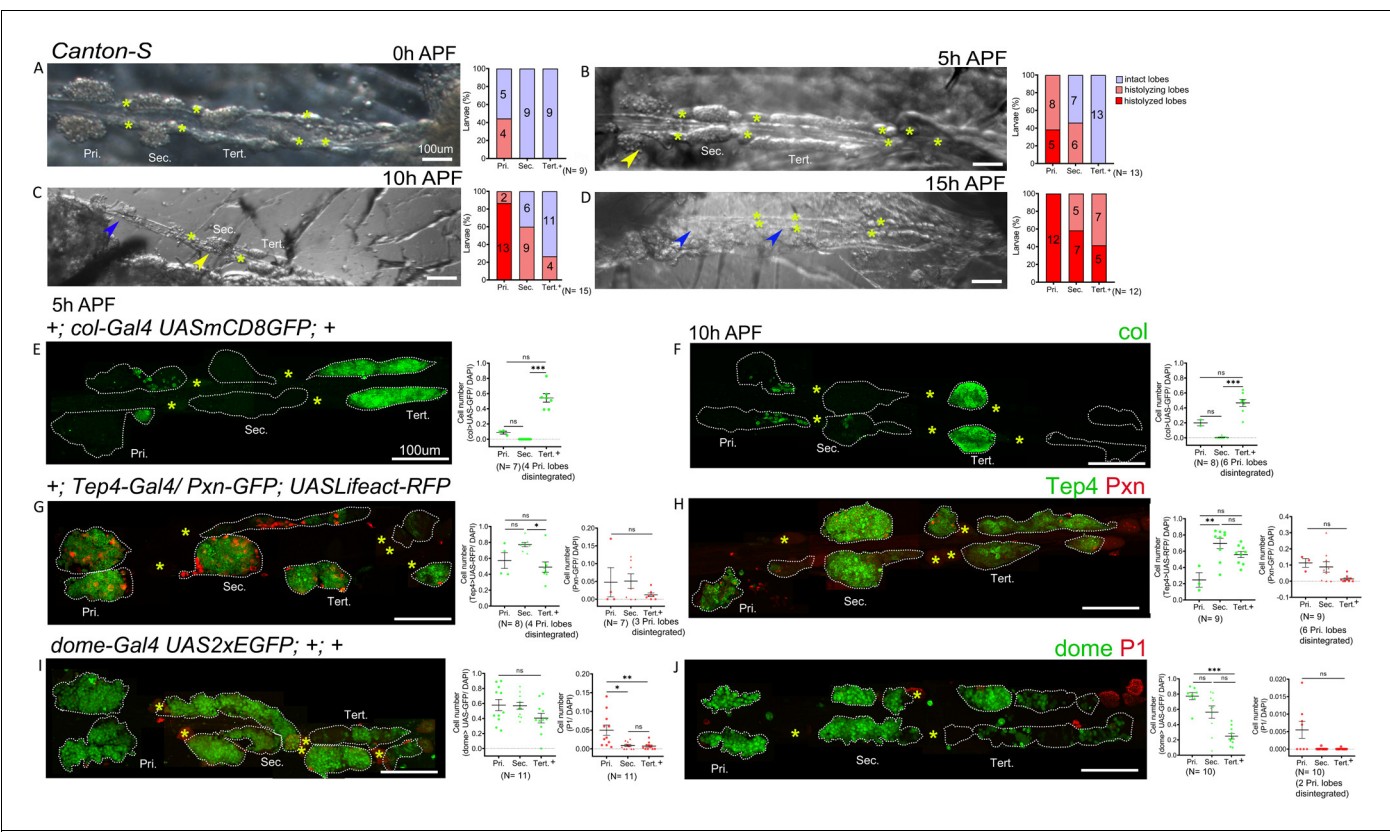

**Figure 3.** Gene expression analysis in the pupal lymph gland. (A–D) Whole lymph gland hemi-dissected preparations indicating lobe histolysis during pupal development at 0 hr after pupa formation (APF), 5 hr APF, 10 hr APF, and 15 hr APF. (A–D) Yellow arrowheads indicate histolyzing lobes and blue arrowheads indicate histolyzed lobes. (E and F) c*ol-Gal4,UASmCD8-GFP* (green) is expressed in the posterior signaling center (PSC) and tertiary lobes during 5 hr APF, 10 hr APF. (G and H) 5 hr APF and 10 hr APF lymph glands express *Tep4-Gal4> UASlifeact-RFP* (green, pseudo color) in the primary lobes and posterior lobes; very few cells express *Pxn-GFP* (red pseudo color) in the primary and posterior lobes. (I and J) 5 hr APF, 10 hr APF lymph glands express *dome-Gal4,UAS2xEGFP* (green) in the primary lobes and posterior lobes, very few cells express P1 (red). Quantification panels indicate ratios of col[+]/DAPI[+] (E and F) or Tep4[+]/DAPI[+] or Pxn[+]/DAPI[+] (G and H) and dome[+]/DAPI[+] or P1[+]/DAPI[+] (I and J) at the indicated time points. (E–J) Kruskal–Wallis test was used for statistical analysis. *p<0.05, **p<0.01, ***p<0.001, ns: nonsignificant, and error bars represent SEM. (A–J) Yellow asterisks indicate pericardial cells, and (E–J) nuclei were stained with DAPI (not shown for clarity). Pri. indicates primary lobes, Sec. indicates secondary lobes, and Tert. indicates tertiary lobes. (A–J) Scale bar: 100 μm.

The online version of this article includes the following source data for figure 3:

**Source data 1.** Numerical data plotted and statistical analysis related to *Figure 3*.

the ECM, showed that it is strongly expressed throughout the posterior lobes and at lower level in the MZ of the anterior lobes (*Figure 2E*). Using an endogenously GFP-tagged version for Apontic (Apt) that acts a transcription factor and negative regulator of JAK-STAT signaling, we found that it is expressed in the posterior lobes, in the heart tube, and at lower levels in the MZ (*Figure 2F*). Similarly, a GFP-5-hydroxytryptamine 1B (5-HT1B) fusion showed that this serotonin receptor is expressed in the posterior lobes and in the MZ (*Figure 2G*). GFP or V5 tagged versions of the guidance molecule Netrin-B (NetB) revealed that it is not only expressed throughout the cardiac tube but also in patches of cells in the posterior lobes, especially in the tertiary lobes, but not in the anterior lobes except for the PSC (*Figure 2H* and *Figure 2—figure supplement 1B,D*). A GFP fusion with Tsp42Ee revealed that this Tetraspanin protein is strongly expressed in the posterior lobes, in particular in the tertiary lobes (*Figure 2I*). Furthermore, we found that the LDL receptor family member CG6024 was expressed in the tertiary lobes and in the PSC, as well as in the pericardial cells (*Figure 2J* and *Figure 2—figure supplement 1C*). Actually, two candidates (the Nephrin homolog Sticks and stones [Sns] and the fatty acid elongase CG31522) were specifically expressed in pericardial cells (*Figure 2K* and *Figure 2—figure supplement 1E*), indicating that contamination by these cells could contribute to the apparent overexpression of some genes in posterior lobe RNA-seq samples. Finally, immunostaining against Ultrabithorax (Ubx) showed that this HOX transcription factor is expressed in some pericardial and heart cells but only in the tertiary lobes within the lymph gland (*Figure 2L*). All together, these data indicate that the lymph gland prohemocytes are more heterogenous in terms of gene expression than previously thought.

In parallel, we used the G-trace system (*Evans et al., 2009*) to assess whether different Gal4 whose expression is under the control of putative enhancers of genes overexpressed in the posterior lobes can drive gene expression in specific territories of the lymph gland, and especially in progenitors and/or in the posterior lobes. Accordingly, we identified three Gal4 lines placed under the control of *dlp* regulatory regions that drive expression in part of the tertiary and secondary lobes (*Figure 2M* and *Figure 2—figure supplement 1F,G*). We also identified an enhancer in *col* driving expression in the MZ as well as in most cells of the posterior lobes (*Figure 2N*), contrary to the classically used *pcol85* lines, which is restricted to the PSC and part of the tertiary lobes (*Figure 1B*; *Benmimoun et al., 2015*). Importantly we found that the *Ubx(M3)-Gal4*, which essentially reproduces *Ubx* expression (*de Navas et al., 2006*), is expressed throughout the posterior lobes but not in the anterior lobes (*Figure 2O*), indicating that the anterior and posterior lobes emerge from distinct territories. To the best of our knowledge, this is the first posterior lobe-specific driver identified and it could be very useful to manipulate gene expression in these cells.

## Heterogeneity in gene expression is maintained in the pupal lymph gland

Previous studies indicated that lymph gland lobes histolyze during the course of pupal development (*Lanot et al., 2001*; *Grigorian et al., 2011*). Secondary lobes histolyze by approximately 8 hr (hours) APF (after pupa formation) (*Grigorian et al., 2011*). However, the analysis did not cover the entire lymph gland or analyze progenitor marker expression. For a more comprehensive description of the fate of posterior lobes, we analyzed the whole lymph gland in hemi-dissected pupal preparations at different time points. Expectedly, at 0 hr APF lymph gland primary lobes begin histolyzing and posterior lobes are maintained intact. At 5 hr APF primary lobes continue to histolyze, while secondary and tertiary lobes begin histolyzing by approximately 5 hr APF and 10 hr APF. By 12–15 hr APF most of the lymph gland is histolyzed (*Figure 3A–D*). Interestingly *col* (*col-Gal4,UASmCD8-GFP*) expression is maintained in the tertiary lobes and both *Tep4* (*Tep4-Gal4,UASLifeact-RFP*) and *dome* (*dome-Gal4,UAS2xEGFP*) are expressed in most cells of the secondary and tertiary lobes at 5 hr and 10 hr APF (*Figure 3E–J*), indicating that progenitors could be maintained during pupal development. Consistent with this hypothesis, we found few differentiated cells at these time points in the posterior lobes as marked by *Pxn-GFP* or P1 (*Figure 3G–J*). Thus, our analyses show that PPs are maintained up to at least 10 hr of pupal development.

## An anterior–posterior graded response to immune stress

*Drosophila* blood cells respond to immune stress caused by bacterial infections and wasp parasitism (*Lanot et al., 2001*; *Sorrentino et al., 2002*; *Crozatier et al., 2004*; *Khadilkar et al., 2017b*;

*Louradour et al., 2017*; *Sinenko et al., 2011*). While all progenitors are assumed to respond uniformly to a given stress, especially to systemic cues, our findings that PPs differ in gene expression led us to hypothesize that this may reflect in the ability to maintain progenitors or differentiate upon immune challenge. Interestingly, infection with the Gram-negative bacteria *E. coli* promoted differentiation in the primary lobes as reported earlier (*Khadilkar et al., 2017b*), but not in the posterior lobes (*Figure 4—figure supplement 1A,B*). *E. coli* has been used extensively to study the innate cellular immune response; however, *E. coli* does not infect *Drosophila* naturally (*Neyen et al., 2014*). *Drosophila* are natural host to parasitoid wasp and the hemocyte immune response is well studied in this context (*Banerjee et al., 2019*). Hence, we tested the sensitivity of the progenitor pools to parasitism by the specialist parasitoid wasp *Leptopilina boulardi*.

Egg deposition by *L. boulardi* activates the humoral and cellular arms of immunity, leading to the production of lamellocytes, notably by the lymph gland, that encapsulate the wasp egg. The response to wasp infestation is well characterized for the primary lobes but data for the posterior lobes are limited (*Lanot et al., 2001*). Of note too, the timing of response seems variable from lab to lab, this could partly be due to the genetic background that is used or the temperature at which the experiment is performed. For instance, *Lanot et al., 2001* observed lamellocyte formation from 10 hr post-parasitism in the primary lobes, while *Sorrentino et al., 2002* reported that lamellocyte production in the primary lobes begins around 51 hr post-parasitism and intensifies by 75 hr. Thus, to assess lamellocyte differentiation and score for morphological changes in the posterior lobe progenitors, if any, we analyzed lymph glands of larvae at 3 days (approximately 75 hr) post-parasitism.

Phalloidin staining or Misshapen (using the *MSNF9-mCherry* reporter; *Tokusumi et al., 2009*) expression were used to detect lamellocytes. To account for the inter-individual variation in the extent of the response, we quantified the phenotypes based on the following broad classification – (a) Strong: lobes completely histolyzed and hence lamellocytes absent or few remnant cells attached to the cardiac tube, (b) Medium: few lamellocytes with histolyzing lobes that show uneven or discontinuous boundaries with loose packing of cells, (c) Mild: no lamellocytes but some cells in the lobe fuse or coalesce in groups with disrupted cell-cell boundaries, and (d) Unaffected: lobes with undisturbed morphology, intact cell–cell connections, and no lamellocytes.

Analysis of the lymph gland post-parasitism revealed differences in the response of progenitors from anterior to posterior. In *Canton-S* (wild-type) background, 93.9% (46/49) of infested larvae showed some response in the primary lobes: 6.1% (3/49) had strong phenotypes, 85.7% (42/49) medium, and 2% (1/49) mild. Only 6.1% (3/49) of anterior lobes seemed unaffected. In contrast, 36.7% (18/49) of the larvae showed no effect on posterior lobes. None of the larvae had strong phenotypes in the posterior lobes and lobes were maintained intact. In addition, secondary and tertiary lobes showed only medium, mild phenotypes such as absence of clear cell–cell boundaries with occasional lamellocytes. 27.08% (13/48) had medium phenotypes in the secondary lobes and 10.41% (5/48) had medium phenotypes in the tertiary lobes. 18.75% (9/48) showed mild phenotypes in the secondary and the tertiary lobes. (*Figure 4A,B*). However, as reported before (*Lanot et al., 2001*; *Sorrentino et al., 2002*; *Crozatier et al., 2004*), posterior lobe size increased upon infestation, indicating that these lobes were indeed responding to the infection.

To ensure that lamellocyte differentiation in the posterior lobes had not occurred earlier, we analyzed lymph glands at day 2 post-parasitism. Again, we found lamellocytes in the anterior but not posterior lobes (*Figure 4—figure supplement 1C*). This suggests that lamellocyte differentiation begins first in the anterior and that primary lobe cells are released into circulation. Furthermore, we analyzed *MSNF9-mCherry* larvae at day 3 post-parasitism and observed a similar pattern as in *Canton-S*, with frequent histolysis in anterior (13/17) but not in posterior lobes (1/17 for secondary lobes and 0/17 for tertiary lobes) (*Figure 4C,D*). Also, the induction of *MSNF9-mCherry* was much stronger in the anterior lobes while most *MSNF9-mcherry*[+] cells present in the posterior lobes remained round and did not exhibit lamellocyte-like morphology, indicating a weaker response of PP (*Figure 4C,D*).

As the PSC is required for the (systemic) response to wasp parasitism and in particular for lamellocyte differentiation in the anterior lobes (*Benmimoun et al., 2015*; *Crozatier et al., 2004*; *Sinenko et al., 2011*), it may contribute to the differential response between AP and PP. To obtain PSC-less lymph glands we overexpressed the pro-apoptotic gene *reaper* using *Antp-Gal4* (*Figure 4—figure supplement 2A,B*; *Benmimoun et al., 2015*). Consistent with previous findings, we find that ablating the PSC prevents lamellocyte differentiation in the primary lobes and we did not

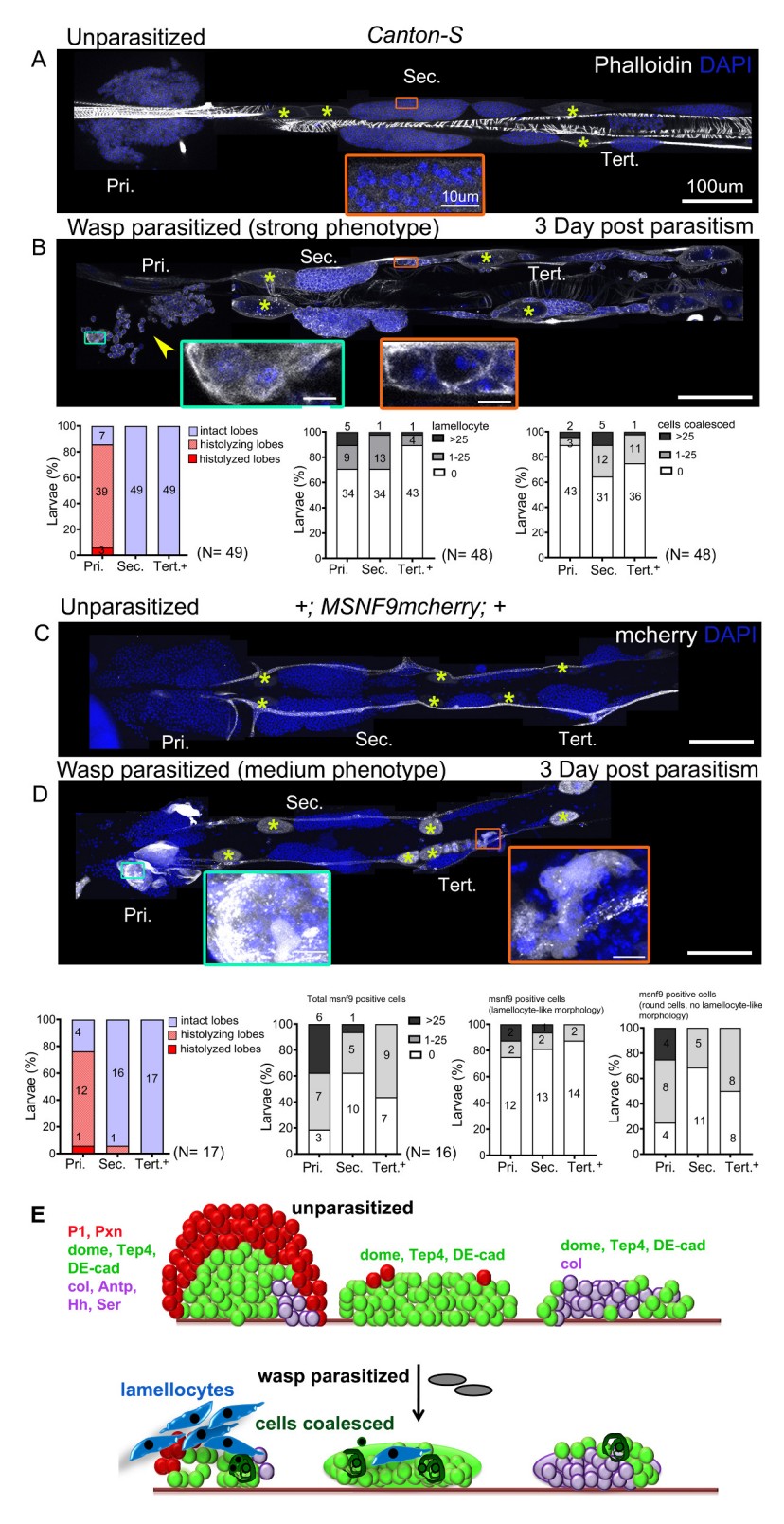

**Figure 4.** Lymph gland progenitor response to wasp parasitism. (**A and B**) *Canton-S* wild-type strain lymph gland lobes were analyzed 3 days post-parasitism by *L. boulardi*. Age-matched unparasitized larvae were used as controls. Phalloidin (white) marks actin and was used for identifying lamellocytes as well as changes in cell morphology. (**B**) Yellow arrowhead indicates disintegrating primary lobes, green inset shows lamellocyte formation in the primary lobes, and orange inset displays compromised cell boundaries in the secondary lobes. (**C and D**) *MSNF9-mcherry* larvae were

*Figure 4 continued on next page*

*Figure 4 continued*

analyzed for lamellocyte formation 3 days post-parasitism. Green and orange insets show lamellocytes in the primary and the posterior lobes, respectively. For quantifications, nuclei showing lamellocyte morphology (phalloidin) or lamellocyte marker expression (MSNF9-mCherry, white pseudo color) were manually counted. Graphs indicate quantification for the cellular phenotypes: premature histolysis of lymph gland lobes, lamellocyte induction, or compromised cell boundaries. Pri. indicates primary lobes, Sec. indicates secondary lobes, and Tert. indicates tertiary lobes. Yellow asterisks indicate pericardial cells. (E) Schematic representation of the lymph gland response to wasp parasitism. (A–D) Scale bars: 100 µm or 10 µm in inset.

The online version of this article includes the following source data and figure supplement(s) for figure 4:

**Source data 1.** Numerical data plotted and statistical analysis related to *Figure 4*.
**Figure supplement 1.** Posterior progenitors (PP) do not differentiate in response to *E. coli* infection and wasp parasitism.
**Figure supplement 1—source data 1.** Numerical data plotted and statistical analysis related to *Figure 4—figure supplement 1*.
**Figure supplement 2.** Posterior signaling center (PSC)-less lymph glands do not differentiate in response to wasp parasitism.
**Figure supplement 2—source data 1.** Numerical data plotted and statistical analysis related to *Figure 4—figure supplement 2*.

observe lamellocyte induction in the posterior lobes in parasitized conditions (*Figure 4—figure supplement 2E–H*). Interestingly, AP of PSC-less larvae did not behave as PP of wild-type larvae as they did not exhibit cell coalescence, a frequent PP response to wasp parasitism. Our analysis suggests that the PSC is not responsible for the differential response between AP and PP. In particular, it is not required to prevent lamellocyte induction in the PP. Intriguingly though, PSC ablation prevented PP cell coalescence, indicating that it could be involved in PP response to parasitism.

## Upregulation of STAT activity in PPs upon wasp parasitism

Our analysis indicates that PP resist differentiation following immune challenge by bacteria (*Figure 4—figure supplement 1A,B*) or wasp (*Figure 4—figure supplement 1C*, *Figure 4E*). To understand the mechanisms that regulate maintenance of PP in parasitized larvae, we analyzed signaling pathways that are implicated in regulating lamellocyte differentiation. It was shown that Toll/NF-kB activation occurs in the AP and in the PSC to promote lamellocyte fate upon wasp infestation (*Gueguen et al., 2013*; *Louradour et al., 2017*). Using the *Drosomycin-GFP (Drs-GFP)* reporter line (*Ferrandon et al., 1998*), we observed activation of Toll/NF-kB signaling in response to wasp parasitism in the AP as reported previously (*Gueguen et al., 2013*; *Louradour et al., 2017*), but also in the PP, although PP show little lamellocyte differentiation (*Figure 5A,B*). These results thus indicate that the differential response of the PP to wasp infestation between anterior and posterior lobes is unlikely to involve the Toll/NF-kB pathway.

Wasp parasitism was found to inhibit Notch activity, resulting in fewer crystal cells and increased lamellocyte differentiation in the AP (*Small et al., 2014*). We used the Notch responsive element (*NRE-GFP*) line to assess Notch activity (*Saj et al., 2010*). In response to wasp infestation primary lobes show reduced Notch activity as reported (*Small et al., 2014*). However, PP barely express this reporter in control conditions, and upon infestation we observed few *NRE-GFP*-expressing cells in the secondary lobes but not in the tertiary lobes (*Figure 5C,D*). This suggests that there is no major involvement of the Notch pathway in PP upon wasp parasitism.

Downregulation of JAK-STAT signaling in the MZ following wasp parasitism is essential for the differentiation of blood cell progenitors into lamellocytes in the primary lobes (*Makki et al., 2010*). In contrast, wasp parasitism leads to an activation of JAK-STAT signaling in circulating hemocytes (and in somatic muscles) (*Yang et al., 2015*) and unrestrained activation of the JAK-STAT pathway in circulating hemocytes promotes lamellocyte differentiation (*Bazzi et al., 2018*). Yet, the role of JAK-STAT signaling in lamellocyte differentiation in the posterior lobes has not been assessed. Hence, we investigated the status of JAK-STAT signaling in the whole lymph gland to gain potential insight into the molecular basis for differential regulation of lamellocyte formation from anterior to posterior.

Accordingly, we used the 10xSTAT-GFP reporter (*Bach et al., 2007*) to analyze STAT92E activation in time-matched control (unparasitized) and parasitized larvae at day 2, 3, or 4 post-parasitism. In control larvae, 10xSTAT-GFP expression was essentially restricted to the posterior part of the anterior lobes at 120 hr (2 days post-parasitism) and 144 hr (3 days post-parasitism) AEL (*Figure 5—figure supplement 1A,C*). However, it was barely detectable at later stage (4 days post-parasitism/168 hr AEL) (*Figure 5E*). In contrast, we observed consistent 10xSTAT-GFP expression throughout

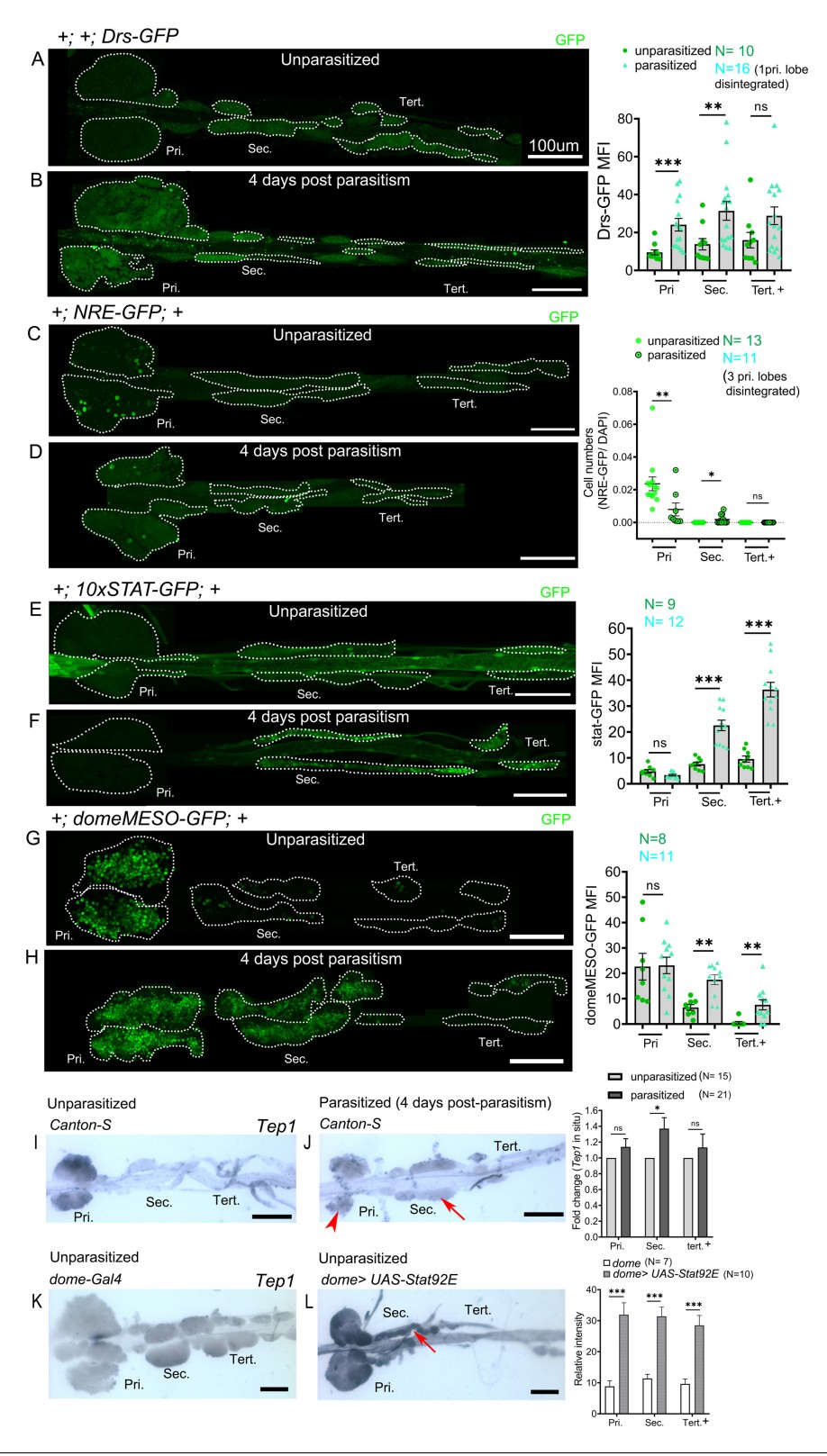

**Figure 5.** Posterior progenitors activate STAT in response to wasp parasitism. (**A–H**) Toll/NF-kB or Notch or JAK-STAT signaling reporter activity (green) in the lymph gland lobes analyzed in age-matched unparasitized and 4 days post-parasitism larvae. (**A and B**) Toll/NF-kB reporter activity monitored using *Drosomycin*-GFP (*Drs-GFP*) expression. Graph represents mean fluorescence intensity (MFI) for *Drs-GFP*. Mann–Whitney nonparametric test was used for statistical analysis. (**C and D**) Notch activity monitored using Notch responsive element (*NRE-GFP*) reporter activity. *Figure 5 continued on next page*

*Figure 5 continued*

Graph represents ratio of NRE-GFP$^+$/DAPI$^+$ cells. Mann–Whitney nonparametric test was used for statistical analysis. (**E–H**) JAK-STAT activity monitored using *10xSTAT-GFP* (**E** and **F**) or *domeMESO-GFP* (**G** and **H**) reporter. Graph represents MFI for 10xSTAT-GFP or domeMESO-GFP. Mann–Whitney nonparametric test was used for statistical analysis. (**I** and **J**) RNA in situ hybridization shows that *Tep1* expression is induced in the secondary lobes (red arrow) post-parasitism as compared to unparasitized larval lymph glands. (**J**) Red arrowhead indicates disintegrating primary lobes. (**K** and **L**) Overexpression of *Stat92E* leads to transcriptional induction of *Tep1* (red arrow) in unparasitized conditions. Student's t-test was used for statistical analysis. Error bars represent SEM. *p<0.05, **p<0.01, ***p<0.001, and ns indicates nonsignificant. Pri. indicates primary lobes, Sec. indicates secondary lobes, and Tert. indicates tertiary lobes. (**A–L**) Scale bars: 100 µm.

The online version of this article includes the following source data and figure supplement(s) for figure 5:

**Source data 1.** Numerical data plotted and statistical analysis related to *Figure 5*.

**Figure supplement 1.** JAK-STAT reporter activity in response to wasp parasitism.

**Figure supplement 1—source data 1.** Numerical data plotted and statistical analysis related to *Figure 5—figure supplement 1*.

**Figure supplement 2.** *10xSTAT-GFP* reporter activity in response to bacterial infection and wasp egg mimic.

**Figure supplement 2—source data 1.** Numerical data plotted and statistical analysis related to *Figure 5—figure supplement 2*.

the posterior lobes from 120 hr to 168 hr AEL in parasitized larvae. Consistent with previous report (*Makki et al., 2010*), STAT92E activity was repressed in the anterior lobes following *L. boulardi* infection, as judged by the quasi absence of 10xSTAT-GFP expression in parasitized larvae (*Figure 5—figure supplements 1B,D* and *Figure 5F*). Strikingly though, STAT92E activity was not repressed in the posterior lobes of parasitized larvae but was even highly upregulated at days 3 and 4 in comparison to unparasitized age-matched control lymph glands (*Figure 5—figure supplements 1C,D* and *Figure 5E, F*).

However, posterior lobes of parasitized 10xSTAT-GFP larvae appeared thinner than parasitized posterior lobes of wild-type larvae. To rule out any anomaly due to this genetic background and confirm STAT92E activation, we analyzed *domeMESO-GFP*, another reporter of activated STAT (*Hombría et al., 2005*; *Louradour et al., 2017*). The 10xSTAT-GFP reporter consists of five tandem repeats of a 441 bp fragment from the intronic region of *Socs36E* with two STAT92E binding sites (*Bach et al., 2007*), whereas the domeMESO construct contains a 2.8 kb genomic fragment spanning part of the first exon and most of the first intron of *dome* (*Hombría et al., 2005*). Previous studies report the downregulation of *domeMESO*-lacZ 30 hr post-parasitism in the primary lobe (*Makki et al., 2010*). We analyzed *domeMESO*-GFP reporter 2 days and 4 days post-parasitism but surprisingly we did not observe reduction in GFP signal in the primary lobes at any of these time points (*Figure 5—figure supplements 1E, F* and *Figure 5G, H*). Persistent signal post-parasitism could be due to GFP perdurance. Nevertheless, 2 days post-parasitism secondary lobes showed significantly high GFP$^+$ cells and at 4 days post-parasitism secondary as well as tertiary lobes had increased GFP-expressing cells, although induction of GFP is lower in tertiary lobes as compared to the secondary lobes (*Figure 5—figure supplements 1E,F* and *Figure 5G, H*).

To gain further evidence for increased activity of STAT92E in the posterior lobes of parasitized larvae, we also assessed the expression of the STAT target *Tep1*. Indeed JAK overactivation as well as *L. boulardi* infection were shown to induce the expression of this opsonin (*Lagueux et al., 2000*; *Wertheim et al., 2005*; *Salazar-Jaramillo et al., 2017*). RNA in situ hybridization revealed a basal level expression of *Tep1* in the primary and posterior lobes in the wild-type unparasitized larval lymph glands (*Figure 5I*). Upon parasitism, *Tep1* expression increased in the secondary lobes but not in primary and the tertiary lobes (*Figure 5J*). In addition, in line with the idea that *Tep1* is a target of the JAK/STAT pathway (*Lagueux et al., 2000*), we found that S*tat92E* overexpression in the progenitors induces *Tep1* expression in unparasitized conditions in the anterior as well as in the posterior lobes (*Figure 5K,L*).

The JAK-STAT pathway participates in various kinds of stress responses. Hence the activation of STAT92E in the posterior lobes may not be specific to wasp parasitism but could reflect a generic response to immune stress. To test this hypothesis, we infected larvae with *Pseudomonas entemophila,* a naturally occurring pathogen that induces a systemic immune response (*Vodovar et al., 2005*), or inserted a human hair in the hemocoel, which triggers lamellocyte differentiation (*Lanot et al., 2001*). However, these challenges did not cause activation of the STAT pathway in the lymph gland posterior lobes as assessed with the 10xSTAT-GFP reporter (*Figure 5—figure supplement 2*).

In sum, our results indicate that STAT92E activation in PP is likely a localized immune response to specific systemic cues triggered by wasp parasitism and suggests the existence of different mechanisms for regulating STAT92E in AP and PP.

## STAT92E limits the differentiation of lamellocytes in the PPs

Post-parasitism downregulation of JAK-STAT signaling is implicated in the differentiation of lamellocytes in the primary lobes (*Makki et al., 2010*). Our analysis thus far suggests that high levels of JAK-STAT signaling in the PP underlie their different response to parasitism. To test if high levels of JAK-STAT signaling inhibit lamellocyte induction in the PP, we knocked-down *Stat92E by RNAi* in the lymph gland. The knockdown of *Stat92E* in the progenitors using *dome-Gal4* (or *tep4-GAL4*) at 25°C did not affect the survival of control larvae but caused lethality in parasitized larvae (*Figure 6— figure supplement 1A,B*), indicating STAT92E activation could be an essential part of the immune response. However, we obtained parasitized escapers when the larvae were raised at 21°C. In these conditions, STAT92E knockdown larvae exhibited stronger responses than controls in the anterior and posterior lobes, as judged by lamellocyte differentiation and presence of fused/coalesced cells (*Figure 6A–D*). Notably, post-parasitism, 60% (6/10) of STAT92E knockdown larvae showed medium phenotypes as compared to 20% (2/10) in parasitized *dome-Gal4* larvae. Additionally, the posterior lobes showed a decrease in progenitors as observed by reduced expression of *dome>GFP* (*Figure 6C,D*), in agreement with our hypothesis that STAT92E blocks PP differentiation post-parasitism.

As described above, the *Ubx(M3)-Gal4* is expressed in the PP but not in the primary lobes. Therefore, we took advantage of this driver to specifically knockdown *Stat92E* in the PP. Upon parasitism, more larvae showed lamellocyte induction in the posterior lobes in STAT92E knockdown conditions than in controls: 93% (13/14) of *Ubx(M3)-Gal4>STAT92E RNAi* larvae show medium phenotype versus 20% (2/10) in *Ubx(M3)-Gal4* controls. Conversely, only 7% (1/14) of *Stat92E* knockdown larvae displayed no phenotype, as compared to 50% (5/10) of control larvae. Additionally, we observed that many PP differentiate to plasmatocytes upon knockdown of *Stat92E* (*Figure 6E–H*). Although *Ubx(M3)-GAL4* is not expressed any more in the secondary lobes of third instar larvae (see *Figure 2O*), these data indicate that *STAT92E* activity is required either at an earlier stage or in a subset of tertiary lobes cells to prevent PP differentiation in response to parasitism. Of note too, in unparasitized conditions, STAT92E knockdown using *dome-Gal4* or *Ubx(M3)-Gal4* does not trigger lamellocyte or plasmatocyte differentiation (*Figure 6—figure supplement 1C,D,G–J*) and *dome>GFP* expressing progenitors are maintained (*Figure 6—figure supplement 1E,F*).

These results strongly suggest that JAK-STAT signaling is essential for maintaining the progenitor pool and restricting lamellocyte differentiation post-parasitism. Our analyses reveal functional compartmentalization in the lymph gland progenitor pool which responds differentially to immune stress along the anterior–posterior axis. To unravel mechanisms regulating this compartmentalization we examined the status of key JAK-STAT pathway components in the whole lymph gland.

## Differential regulation of the JAK-STAT inhibitor *eye transformer/latran* in lymph gland progenitors

Three ligands, Unpaired (Upd), Upd2, and Upd3 bind to the receptor Dome and trigger JAK-STAT signaling but only Upd3 is expressed in the lymph gland (*Makki et al., 2010*), both in the anterior and the posterior lobes (*Supplementary file 1*). Moreover *upd3* is required for JAK-STAT activation in the anterior lobes and its expression is downregulated 4 hr after wasp infestation (*Makki et al., 2010*). In agreement with a previous report (*Jung et al., 2005*), we found that the reporter line *upd3-GAL4* (*Agaisse et al., 2003*) is expressed in the MZ of the anterior lobes, as reported for *upd3* transcript (*Makki et al., 2010*), but also in the posterior lobes. Upon wasp parasitism its expression level was reduced in primary lobes but not in the posterior lobes (*Figure 7A,B*), even though they exhibit increased STAT92E activity at that time (see above). We also assessed the expression of the JAK-STAT pathway receptor *dome* using the *dome-GAL4* reporter (*Bourbon et al., 2002*; *Jung et al., 2005*). *dome-Gal4* is expressed in AP and PP in unparasitized larvae and, consistent with the previous report (*Krzemień et al., 2007*), we observed that it is downregulated in the AP following parasitism (*Figure 7C,D*). Yet, no change was detectable in the PP. To confirm these results, we also directly assessed *upd3* and *dome* expression using RNA in situ. We found that both transcripts

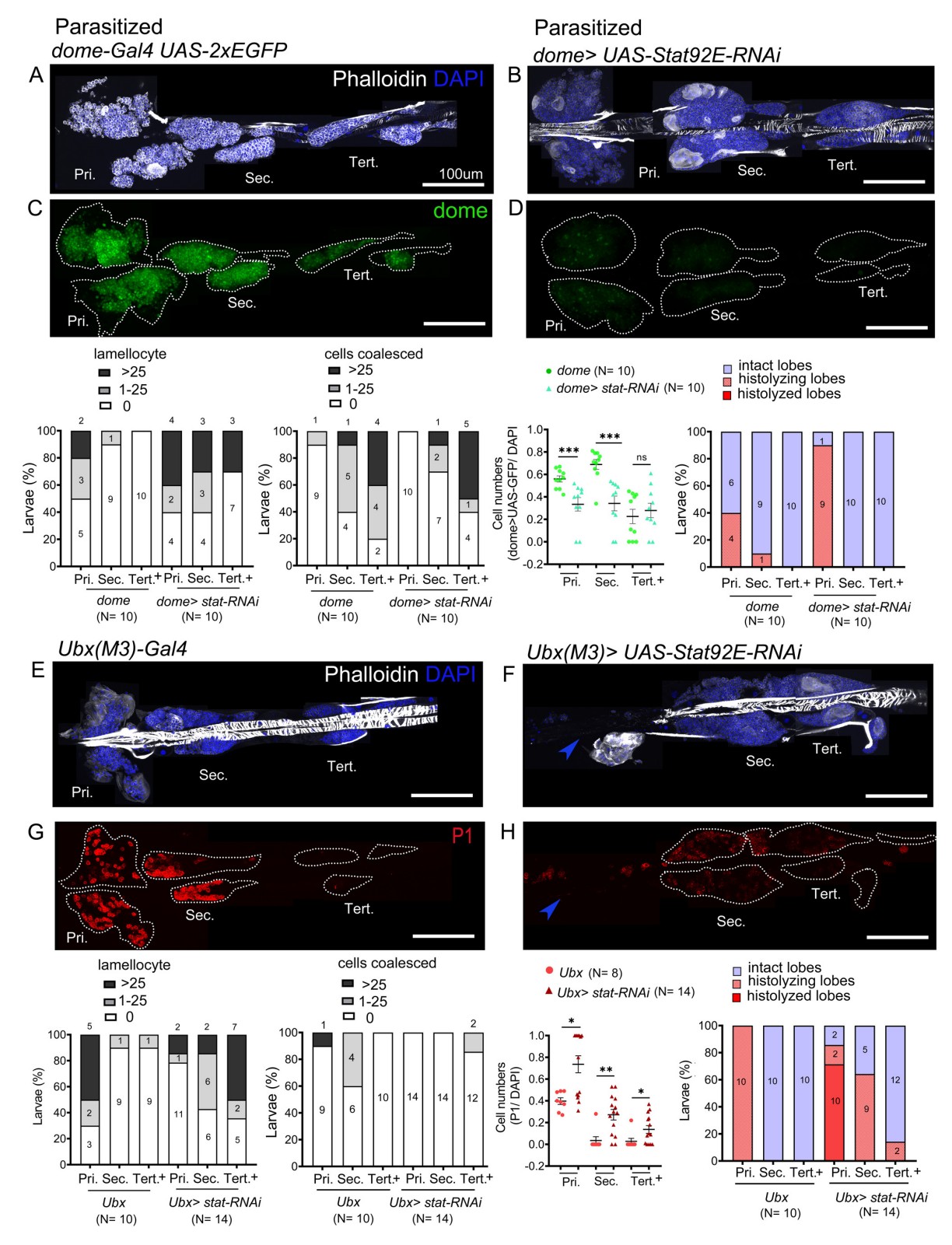

**Figure 6.** STAT limits lamellocyte differentiation in the posterior lobes. Lymph glands were analyzed 4 days post-parasitism, and age-matched parasitized larvae were used for comparison. (A–H) Phenotypes for targeted knockdown of S*tat92E* in the progenitor population are shown. (A–D) S*tat92E* knockdown in the progenitors is associated with massive lamellocyte differentiation in the posterior lobes as revealed by phalloidin staining (white) (A and B) and a reduction in the progenitor pool as indicated by *dome-Gal4,UAS-2xEGFP* (green) expression (C and D) in parasitized larvae. (E–

*Figure 6 continued on next page*

*Figure 6 continued*

H) Posterior progenitors (PP) specific knockdown of S*tat92E* induces lamellocyte and plasmatocyte differentiation in the posterior lobes following parasitism. Graphs indicate quantification for the phenotypes-lamellocyte induction, compromised cell boundaries, dome⁺ or P1⁺ relative cell number, and lobe histolysis. (F and H) Blue arrowheads indicate histolyzed lobes. Mann–Whitney nonparametric test was used for statistical analysis of dome and P1 and error bars represent SEM. *p<0.05, **p<0.01, ***p<0.001, and ns indicates nonsignificant. Pri. indicates primary lobes, Sec. indicates secondary lobes, and Tert. indicates tertiary lobes. (A–H) Scale bar: 100 µm.

The online version of this article includes the following source data and figure supplement(s) for figure 6:

**Source data 1.** Numerical data plotted and statistical analysis related to *Figure 6*.
**Figure supplement 1.** *Stat92E* is essential for survival post-parasitism.

were downregulated not only in the primary lobes but also in the secondary and tertiary lobes in parasitized larvae as compared to unparasitized controls (*Figure 7—figure supplement 1A–D*). The differences we observe between the Gal4/UAS signals and the endogenous transcripts may reflect the perdurance of the GFP and/or the *Gal4* insertions may not fully recapitulate *upd3* and *dome* transcriptional regulation. Yet, these results strongly suggest that parasitism-induced activation of the JAK-STAT pathway in PP is not due to regulation of *upd3* or *dome* expression and that other controls should operate.

Along that line, we previously showed that the endosomal protein Asrij, a conserved modulator of stem and progenitor maintenance, positively regulates STAT92E activation in the lymph gland (*Sinha et al., 2013*; *Figure 7—figure supplement 2A,B*). Hence, we investigated whether increased STAT92E activation in the PP upon wasp parasitism could be a result of increased Asrij expression. However, Asrij protein levels were significantly reduced in both anterior and posterior lobes upon wasp parasitism (*Figure 7E,F*) suggesting an *asrij*-independent mechanism of STAT92E activation post parasitism. In agreement with this, *asrij* null lymph glands still showed increased 10xSTAT-GFP reporter expression in the posterior lobes following parasitism (*Figure 7—figure supplement 2C, D*).

In addition, we found that Dlp, an ECM component known to promote JAK-STAT signaling in the eye disc and the ovaries by stabilizing Upd (*Hayashi et al., 2012*; *Zhang et al., 2013*), is highly expressed in the posterior lobes (*Figure 2D*). It could thus participate in the differential regulation of the JAK-STAT pathway between AP and PP and/or in the regulation of PP fate. Immunostaining showed no change in Dlp levels post-parasitism and it remained high in posterior lobes (*Figure 7G, H*). Nonetheless, we tested the role of Dlp in the response to parasitism by knocking-down its expression by RNAi using the *e33c-Gal4* that shows widespread expression in the lymph gland lobes (*Harrison et al., 1995*; *Kulkarni et al., 2011*). Interestingly, the knockdown of *dlp* increased the occurrence of mild phenotypes in the posterior lobes as compared to control larvae, whereas it did not enhance the response to parasitism in the anterior lobes (*Figure 7I,J*). This suggests that Dlp could promote JAK-STAT signaling in the posterior lobes, leading to different responses of the AP and PPs.

Eye transformer/Latran is a negative regulator of JAK-STAT signaling (*Kallio et al., 2010*; *Makki et al., 2010*). In the lymph gland primary lobes, downregulation of JAK-STAT signaling by Latran is required for lamellocyte differentiation (*Makki et al., 2010*). RNA in situ analysis of unparasitized lymph gland showed high levels of *latran* in the primary lobes as reported (*Makki et al., 2010*), but also in the secondary lobes and in a part of the tertiary lobes (*Figure 7K*). Interestingly, upon parasitism *latran* expression is reduced in the secondary and the tertiary lobes (*Figure 7L*), indicating that differential pathway activation may be regulated by Latran. To substantiate our results, we perturbed JAK-STAT signaling in the AP and/or PP using different Gal4 drivers expressed widely in the lymph gland (*e33c-Gal4*; *Harrison et al., 1995*; *Kulkarni et al., 2011*) or in (part of) the posterior lobes (*dlp-Gal4* and *UbxM3-Gal4*, see *Figure 2M,O*). We find that knocking down *dlp* using *e33c-Gal4* decreased survival post-parasitism as compared to parasitized Gal4 controls (*Figure 7M*). Moreover, attenuating JAK-STAT signaling by knocking down *stat92E* in the PP using *dlp-Gal4* or *UbxM3-Gal4* also leads to reduced survival (*Figure 7N,O*). These data strongly suggest that increased activation of JAK-STAT signaling in the PP or PP subsets maintains progenitors and improves survival rate in response to wasp parasitism.

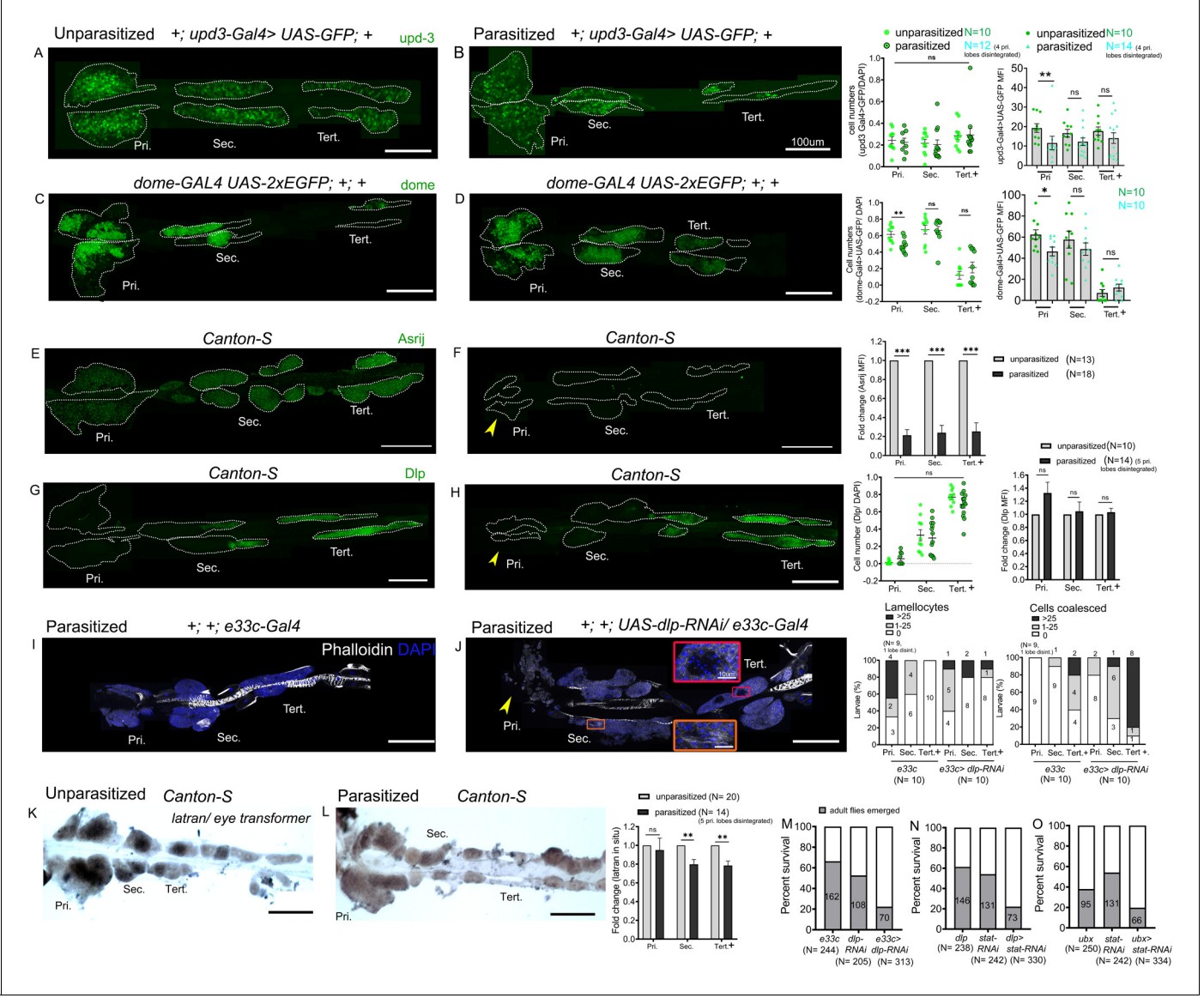

**Figure 7.** Differential regulation of the JAK-STAT inhibitor *latran* in anterior progenitors (AP) and posterior progenitors (PP) following parasitism. Lymph glands were analyzed 4 days post-parasitism, and age-matched unparasitized larvae were used for analysis in each case. (**A and B**) *upd3-Gal4> GFP* (green) expression reduces in the primary lobe and remains unchanged in the posterior lobes in parasitized conditions. *dome-Gal4 >2* xEGFP (green) expression in unparasitized (**C**) and parasitized (**D**) conditions show reduced expression of *dome* in the primary lobes post-parasitism. Graphs represent ratio of cells positive for each marker or mean fluorescence intensity (MFI) in the respective lobes. Mann–Whitney nonparametric test was used for statistical analysis. (**E and F**) show reduced expression of Asrij (green) in the lymph gland post-parasitism. Graph represents fold-change for MFI in the respective lobes; Student's t-test was used for statistical analysis. Dlp (green) expression remains unchanged in unparasitized (**G**) and parasitized (**H**) conditions. Graphs represent ratio of Dlp+/DAPI+ and MFI in the respective lobes. Mann–Whitney nonparametric test and Student's t-test were used for statistical analysis. (**I and J**) Targeted knockdown of *dlp* leads to lamellocyte formation and affects cell boundaries in the posterior lobes. Phalloidin (white) marks actin and is used for identifying lamellocytes and changes in cell morphology. (**J**) Orange inset shows lamellocyte formation in the secondary lobes and red inset displays compromised cell boundaries in the tertiary lobes. (**K and L**) RNA in situ hybridization for *latran* shows decreased expression in the PP post-parasitism. Graph represents fold-change; Student's t-test is used for statistical analysis. (**M–O**) indicate percentage of survival in JAK-STAT signaling perturbed conditions. (**F, H, and J**) Yellow arrowhead indicates disintegrating primary lobes. Error bars represent SEM. *p<0.05, **p<0.01, ***p<0.001, and ns indicates nonsignificant. Pri. indicates primary lobes, Sec. indicates secondary lobes, and Tert. indicates tertiary lobes. (**A–L**) Scale bar: 100 µm or 10 µm in inset.

The online version of this article includes the following source data and figure supplement(s) for figure 7:

**Source data 1.** Numerical data plotted and statistical analysis related to *Figure 7*.

**Figure supplement 1.** *dome* and *upd3* transcripts are downregulated upon parasitism.

**Figure supplement 1—source data 1.** Numerical data plotted related to *Figure 7—figure supplement 1*.

*Figure 7 continued on next page*

*Figure 7 continued*

**Figure supplement 2.** Asrij-independent upregulation of STAT activity in posterior progenitors following parasitism.

## Discussion

The *Drosophila* lymph gland has been used as a powerful system to study blood cell progenitor maintenance. However, intra-population heterogeneity has not been explored previously. Here we introduce the lymph gland as a model to analyze progenitor heterogeneity at the level of the complete hematopoietic organ by phenotypic marker expression, proliferative capacity, and differentiation potential. We show that posterior lobes acquire cells during the third instar stage of development. The exponential increase in the size of the posterior lobes is accompanied by increased cell proliferation and mitotic activity from the mid-third instar to the wandering larval stages. At the wandering larval stage posterior lobes consist of a significant population of progenitors and are essentially devoid of differentiated cells. PP express progenitor markers like *Tep4, dome, DE-cad* in the secondary lobes and in some cells of the tertiary lobe but very few cells expressing differentiation markers such as Pxn, P1/NimC1, or ProPO are present. Furthermore, while most cells of the posterior lobes express low levels of *col* like the APs, part of the tertiary lobes express high levels of *col*, like the PSC in the anterior lobes. However, we did not observe expression of other PSC markers such as *Antp* or *hh*, suggesting that there is no clear homologue of the PSC/niche in these lobes. While several autonomous factors and local signals emanating from the PSC have been implicated in progenitor maintenance in the anterior lobes (*Banerjee et al., 2019*), how progenitor fate is maintained in the posterior lobes remains largely unknown and certainly deserves further investigation. The identification of genes overexpressed in the PP, such as the transcription factors Apt or the ligand Net-B, paves the way for such investigations.

AP are responsive to systemic cues/long range signals present in the hemolymph. For instance, they are sensitive to nutrient deprivation and olfactory cues, which regulate specific signaling pathways in the anterior lobes under physiological conditions (*Benmimoun et al., 2012*; *Shim et al., 2012*; *Tokusumi et al., 2012*; *Shim et al., 2013*). Olfactory-immune crosstalk also helps in priming larvae to respond rapidly in conditions of immune stress (*Madhwal et al., 2020*). It would be all the more interesting to study whether the same processes operate in PP as we revealed clear differences between AP and PP response to systemic cues elicited by immune challenges. Our finding that *5-HT1B* and *5-HT1A*, which code for serotonin receptors, are expressed at higher levels in the posterior lobes suggests that progenitor fate or function could be controlled by serotonin circulating through the hemolymph or produced by neighboring cells. Indeed, besides its function as a neurotransmitter, serotonin acts as a peripheral hormone and has an immunomodulatory effect on blood cells (*Herr et al., 2017*). Notably, mice deficient for the serotonin receptor 5-HT2B show altered bone marrow composition, with increased granulocyte precursors and reduced immature endothelial precursors (*Launay et al., 2012*). Moreover, phagocytosis is severely impaired in *Drosophila* mutants for serotonin receptors 5-HT1B and 5-HT2B (*Qi et al., 2016*).

Although our transcriptome analysis revealed a strong over-representation of genes associated with ECM organization (such as the laminins LanB1/B2/A, Collagen 4A1, Viking, Tiggrin, Glutactin, or Papilin) among those overexpressed in the primary lobes, the HSPG Dlp and the ECM receptor Dg were overexpressed in the PP. Dlp was shown to regulate PSC size by regulating the response to the Decapentaplegic (Dpp) signaling pathway (*Pennetier et al., 2012*), but its function in the prohemocytes is unknown. Dg binds ECM proteins like Perlecan, whose mutation causes reduced lymph gland growth and premature differentiation of the AP in the primary lobes (*Grigorian et al., 2013*). Similarly, the ECM protein Tiggrin was found to control the progression of intermediate progenitors to mature plasmatocytes (*Zhang and Cadigan, 2017*). *Grigorian et al., 2011* also suggested that hemocytes digest only a small part of the ECM to facilitate their dispersal and most of the ECM is left intact during metamorphosis. Systemic as well as local signals regulating ECM secretion and adhesiveness play an important role in vertebrate bone marrow to regulate blood cell quiescence, maintenance, or egress (*Klamer and Voermans, 2014*; *Gattazzo et al., 2014*; *Zhang et al., 2019*; *Khadilkar et al., 2020*). However, in-depth analysis of the widely dispersed vertebrate hematopoietic compartment is technically challenging. A deeper understanding of the differential expression of ECM components in the *Drosophila* lymph gland and how they regulate signaling in and

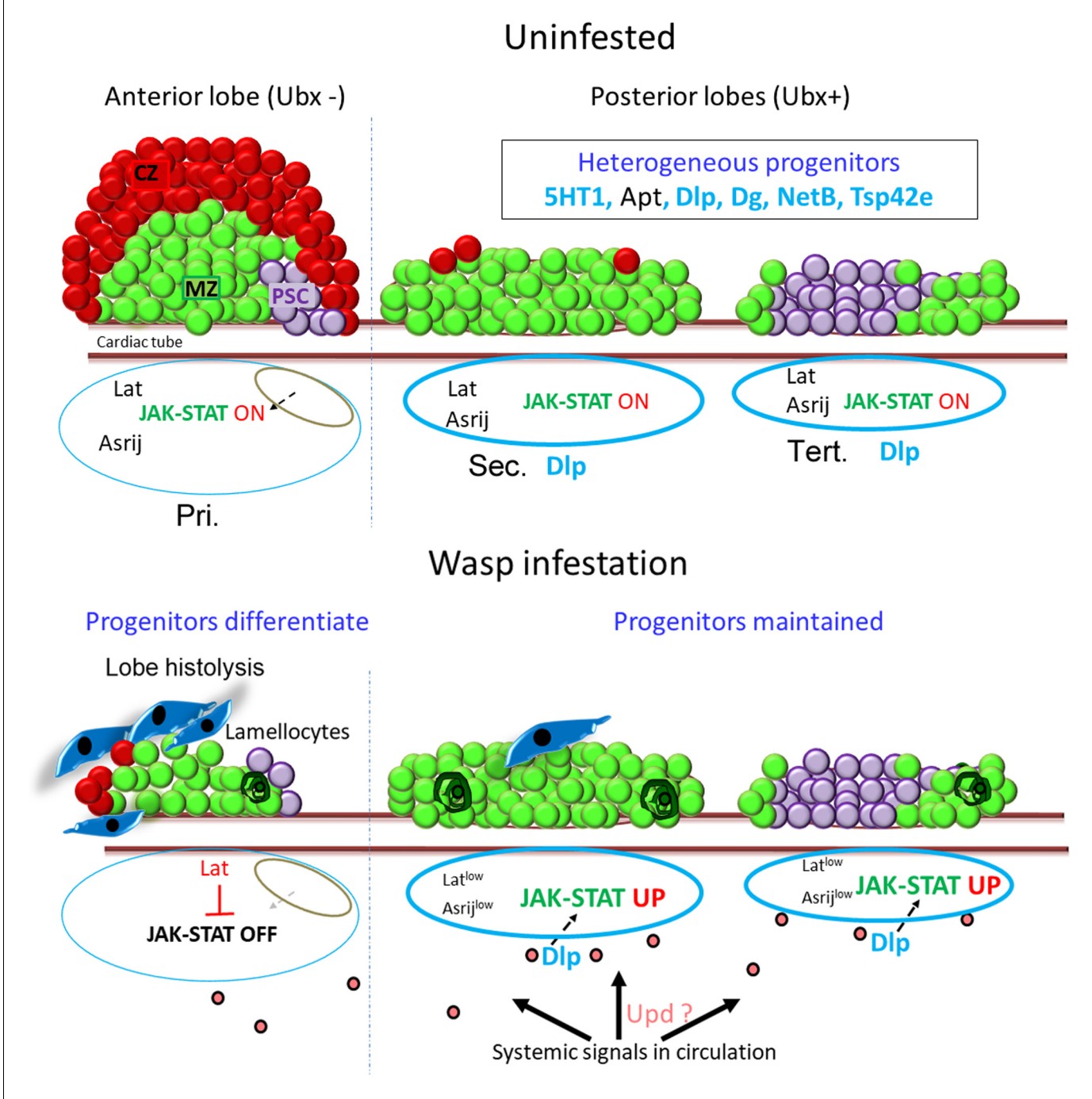

**Figure 8.** Model depicting lymph gland progenitor heterogeneity and response to immune challenge. Schematics showing the entire *Drosophila* third instar (wandering) larval lymph gland. Uninfested – primary lobe is demarcated into the niche/PSC (purple), progenitor/MZ (green), and differentiated/ CZ (red) zones. Posterior lobes harbor a heterogeneous progenitor pool that is not segregated into zones. Posterior progenitor (PP) markers identified from this study are indicated in light blue (extracellular ligands, matrix components, and receptors) or black (transcription factors) font. The status of JAK-STAT signaling is indicated below for each lobe. Upon wasp infestation, JAK-STAT pathway is downregulated in anterior progenitors leading to lamellocyte differentiation (dark blue) and histolysis. Systemic signals lead to production of ligands such as Upd (pink) that may be selectively trapped by extracellular matrix components such as Dlp around the posterior lobes (light blue border), leading to increased JAK-STAT signaling, in an Asrij-independent and Latran-dependent manner. PPs are maintained and few coalesced cells are seen (dark green). Pri. indicates primary lobes, Sec. indicates secondary lobes, and Tert. indicates tertiary lobes. PSC: posterior signaling center; MZ: medullary zone; CZ: cortical zone.

maintenance of progenitors could reveal evolutionary conserved mechanisms that help understand bone marrow hematopoiesis.

Our results bring new insights into the origin and fate of the posterior lobes during development. Posterior lobes flank the dorsal vessel in the abdominal segments (*Jung et al., 2005*; *Banerjee et al., 2019*); however, there is no literature regarding the ontogeny of the posterior lobes. In the embryo the homeotic gene *Ubx* provides positional cues to restrict the formation of the primary lobes to the thoracic segments (*Mandal et al., 2004*). Yet, our transcriptome show that *Ubx* is strongly expressed in the PP and we observe that Ubx is specifically expressed in the tertiary lobes of third instar larval lymph glands. Moreover, lineage tracing analysis with *Ubx(M3)-Gal4* strongly suggests that all the cells of the posterior lobes (and none of the anterior lobes) are derived from the Ubx$^+$ anlage, that is, the embryonic abdominal segments A1–A5 (*Lo et al., 2002*). It would be interesting to study the role of Ubx and the more posterior Hox Abdominal-A (Abd-A) or Abdominal-B (Abd-B) (which are not expressed in the AP or PP) in providing positional cues for seeding the posterior lobes. Likewise several Hox, including Ubx ortholog HoxA7, have been implicated in HSPC development in mammals (*Collins and Thompson, 2018*). Whether they contribute to the emergence of distinct pools of blood cell progenitors during development is still unknown.

Lymph gland lobes histolyze at the onset of metamorphosis (*Lanot et al., 2001*; *Grigorian et al., 2011*; *Makhijani et al., 2011*; *Gold and Brückner, 2015*). A previous study focused on the anterior lobes showed that secondary lobes express Pxn at 4 hr APF but disperse before terminal differentiation (*Grigorian et al., 2011*). We observe that in all the posterior lobes, cells that persist till 10 or 15 hr APF still express progenitor markers and do not undergo terminal differentiation. Pupal and adult blood cells derive from the embryonic and lymph gland lineages (*Holz et al., 2003*). We speculate that under normal conditions PP act as a reserve/long-term pool of progenitors that could have specific functions in larval and/or pupal and adult stages. Along that line, it was proposed that undifferentiated blood cells derived from *col*-expressing posterior lobe hemocytes persist in the adult (*Ghosh et al., 2015*), but a recent study challenged these conclusions and found no evidence for active hematopoiesis during adulthood (*Sanchez Bosch et al., 2019*). However, a detailed lineage tracing analysis for understanding the contribution of AP and PP to pupal and adult stages has not been possible. It is anticipated that the present identification of new progenitor markers and of a Gal4 driver specifically expressed in the posterior lobes should help characterize the fate of lymph gland progenitors in the pupal and adult stages.

Importantly, we show here that the PP exhibit a distinct behavior as compared to AP and our data support the hypothesis that differential regulation of the JAK-STAT pathway underlies this functional compartmentalization. In response to wasp parasitism downregulation of JAK-STAT signaling is required for the differentiation of lamellocytes in the primary lobes (*Makki et al., 2010*; *Louradour et al., 2017*). Our analysis of the lymph gland in its entirety indicates that PP exhibit higher level of JAK-STAT signaling than the anterior lobes and further activate the pathway in response to wasp parasitism. Moreover, the maintenance of high level of STAT92E activity in the PP appears to be required to prevent lamellocyte differentiation in these cells. Interestingly, wasp infestation was also shown to cause JAK-STAT signaling activation in the larval somatic muscle, which is essential for mounting an efficient immune response (*Yang et al., 2015*). Thus, wasp infestation seems to lead to differential regulation of the JAK-STAT pathway in multiple target organs. Notably, sustained JAK-STAT signaling in the PP, by contributing to the expression of complement like factor Tep1, could further activate the humoral arm of immunity. Along that line, our observation that sustained STAT92E activation in PP is important for fly survival following wasp infestation also underlines the biological relevance of PP differential regulation.

Notably, we monitored Toll/NF-kB and Notch signaling to test whether these pathways are involved in regulating differential immune response across the anteroposterior axis. Indeed Toll/NF-kB activation and Notch repression were implicated in lamellocyte production in the anterior lobes (*Gueguen et al., 2013*; *Hao and Jin, 2017*; *Louradour et al., 2017*; *Small et al., 2014*). However, we found that Toll signaling was induced both in the anterior and in the posterior lobes in response to wasp parasitism, and Notch signaling was barely active in the posterior lobes in control as well as infected larvae. Thus, the differential response of AP and PP is unlikely to involve these two pathways. Yet, activation of the Toll reporter *Drs-GFP* in the PP in response to wasp infection further supports the idea that they could contribute to the humoral immune response. Indeed, Toll/NF-kB

signaling is widely implicated in the regulation of AMP production in response to infections (*Lemaitre and Hoffmann, 2007*).

Along the same lines, it is interesting to note that while infestation by the specialist wasp *L. boulardi* induces melanization and lamellocyte differentiation, venom from the generalist *L. heterotoma* is more immune suppressive and causes hemocyte lysis (*Schlenke et al., 2007*). Toll and JAK/STAT pathway genes are some of the most highly expressed genes in the anti-parasite immune response to *L. boulardi* leading to Tep1 upregulation, which is not seen in *L. heterotoma* infested flies. However, upstream signals that bring about this regulation are not known. The virulence of the parasitoid and the immune response of the host define which species prevails. Several studies show that parasitoid wasps exert a selective pressure on *Drosophila* larvae in order to overcome infection (*Kraaijeveld et al., 2001*; *Leitão et al., 2020*; *McGonigle et al., 2017*). Exposing *Drosophila* larvae to *L. boulardi* and selecting flies that survive parasitism over generations lead to constitutive upregulation of immune-inducible genes (including *Drs*) in such selected populations (*Leitão et al., 2020*). It would be interesting to know whether PP contribute to the heightened immune response in such resistant *Drosophila* populations.

The PSC is critically required for lamellocyte production in the anterior lobes (*Benmimoun et al., 2015*; *Crozatier et al., 2004*) and PSC-derived systemic signals are also essential for the differentiation of lamellocytes in the circulating blood cells (*Sinenko et al., 2011*). Similar systemic signals emanating from the PSC or relayed by circulating blood cells could contribute to PP response to wasp parasitism as we observed that PSC ablation abrogated cell–cell coalescence in the posterior lobes. In addition to the PSC, signals derived from the differentiated blood cells in the CZ contribute to AP homeostasis (*Mondal et al., 2011*). Hence, further investigations will be required to elucidate whether systemic signals emanating from the PSC, the CZ, or the circulating hemocytes could influence PP fate in physiological or immune conditions.

Although the decrease in *latran* expression that we observe in the PP following wasp parasitism could strengthen JAK-STAT signaling, how this pathway is activated in the PP remains unclear. Integration of external stimuli to produce an appropriate response is essential for maintaining blood cell homeostasis and survival. While wasp parasitism causes a decrease in *upd3* expression in the lymph gland (*Makki et al., 2010*) and our own results, it also triggers Upd3 and Upd2 upregulation in circulating hemocytes, which leads to JAK-STAT signaling activation in the larval muscles (*Yang et al., 2015*) and could contribute to JAK-STAT activation in PP. Besides, Dlp is known to promote JAK-STAT activation by binding Upd cytokines (*Zhang et al., 2013*) and it is expressed at high levels in the PP. Dlp could thus help sequester Upd2 and/or Upd3 from the hemolymph and promote JAK-STAT signaling in the PP. More generally, enhanced expression of the ECM components, like the HSPG Dlp, in the posterior lobes may help integrate long-range cytokine signals leading to local activation of pathways in response to immune stress. Furthermore, the role of signals such as serotonin or developmental events remains to be investigated.

In summary, we show that the *Drosophila* larval hematopoietic organ, the lymph gland, has a heterogeneous pool of progenitors whose maintenance is spatially and temporally regulated and we reveal a previously unexpected role for JAK-STAT signaling in maintaining the PPs in the presence of immune challenge (*Figure 8*). Our analysis underscores the linear arrangement and genesis of lymph gland lobes with the youngest and most immature progenitors at the posterior end. Further we identify additional reagents to drive gene expression in specific subsets of progenitors. This provides unique opportunities for visualization and sophisticated developmental analysis of blood progenitor specification and differentiation within the same animal. The *Drosophila* lymph gland could serve as a potent model to understand blood cell progenitor heterogeneity and our findings pave the way for future investigation aimed at understanding the spatio-temporal regulation of progenitor fate in normal and pathological situations in vivo. Wasp infestation provides refined and relevant interventions, akin to a sensitized genetic background, to highlight the context-dependent roles of signaling and progenitors. Using infestation as a tool we could uncover differences in the progenitor response to systemic signals and evolutionarily conserved signaling pathways that regulate these. Understanding such interactions between the immune system and its stimuli could inform developmental analysis and our work sets the stage for future studies to decipher these.

## Materials and methods

### Fly stocks and genetics

*Drosophila* stocks were maintained under standard rearing conditions at 25°C, unless specified otherwise. *Canton-S* was used as the wild-type reference strain. *dome-Gal4,UAS-2xEGFP, Tep4-Gal4, Ser-Gal4,UAS2xEYFP, Antp-Gal4* (provided by Utpal Banerjee, University of California Los Angeles), *pcol85-Gal4, UASmCD8-GFP* (provided by Michèle Crozatier, University of Toulouse), *MSNF9-mcherry, hhF4f-GFP* (provided by Robert Schulz, University of Notre Dame), *Drosomycin-GFP* (provided by Dominique Ferrandon), *domeMESO-GFP* (provided by Tina Mukherjee, NCBS), *upd3-Gal4, UAS-GFP* (provided by Sveta Chakrabarti, Indian Institute of Science), *UAS-Stat92E-RNAi* (VDRC #43866), *UAS-dlp-RNAi* (NCBS Fly facility) (VDRC #10299), *UAS-Stat92E* (#F000750) (*Bischof et al., 2013*) (Fly ORF, Zurich ORFeome project), *e33c-Gal4, arj⁹/arj⁹,* (*Kulkarni et al., 2011*), Fly-FUCCI (BL55121), Fly-FUCCI (BL55122), *G-trace* (BL28281), *10xSTAT-GFP* (BL26197), *10xSTAT-GFP* (BL26198), *5-HT1B-GFP* (BL60223), *apt-GFP* (BL51550), *CG31522-GFP* (BL64441), *CG6024-GFP* (BL64464), *col (GMR13B08)-Gal4* (BL48546), *dlp (GMR53G07)-Gal4* (BL46041), *dlp (GMR53E05)-Gal4* (BL48196), *dlp (0421-G4)-Gal4* (BL63306), *E(spl)mß-HLH-GFP* (BL65294), *netB-GFP* (BL67644), *netBtm-V5* (BL66880), *sns-GFP* (BL59801), *Tsp42Ee-GFP* (BL51558), *UAS-Lifeact-RFP* (BL58362), *UAS-reaper* (BL5824), *tub-Gal80ᵗˢ* (BL7017), *NRE-GFP* (BL30727), *Ubx(M3)-Gal4* (provided by Ernesto Sanchez-Herrero, CBMSO, University of Madrid). To generate the *Pxn-GFP* reporter line, regulatory region (dm6 chr3L:2629330–2629639) was cloned into the pH-stinger vector (DGRC #1018) and the corresponding transgenic lines were generated by standard P-element-mediated transformation into *w¹¹¹⁸* flies.

### Whole lymph gland sample preparation

Briefly, larvae or staged pupae were washed in 1× PBS, placed dorsal side facing up and pinned at the anterior and posterior ends. Larvae were slit laterally and the cuticle was carefully cut along the edges, loosened and lifted away gently to expose viscera which were then removed. The entire lymph gland, attached to the brain lobes in the anterior and flanking the dorsal vessel which was thus exposed, was washed and fixed in this preparation. Fixed hemi-dissected larvae were transferred to a 96-well dish and processed for staining. For mounting, stained hemi-dissected preparations were transferred to a cover slip dish, the whole lymph gland was carefully separated from rest of the larval cuticle with fine scissors.

### Sample preparation and RNA-sequencing

Lymph glands from *w¹¹¹⁸* female wandering third instar larvae were dissected in ice cold PBS, the anterior and the posterior lobes of the lymph gland were separated (taking care to remove the ring gland which is close to the anterior lobes), transferred to an Eppendorf containing 10 µl of RNAlater (FisherScientific #10564445) and frozen on dry ice. Independent biological triplicates were prepared for each condition. RNA extraction was performed using Arcturus PicoPure RNA kit (ThermoFisher #KIT0204). RNA samples were run on the Agilent Bioanalyzer to verify sample quality. Samples were converted to cDNA using Nugen's Ovation RNA-Seq System (Catalogue # 7102-A01). Libraries were generated using Kapa Biosystems library preparation kit (#KK8201) and multiplexed libraries were sequenced on a 1 × 75 High output flow cell on the NextSeq550 platform (Illumina). Reads were filtered and trimmed to remove adapter-derived or low-quality bases using Trimmomatic and checked again with FASTQC. Illumina reads were aligned to *Drosophila* reference genome (dm6 Ensembl release 70) with Hisat2. Read counts were generated for each annotated gene using HTSeq-Count. RPKM (Reads Per Kilobase of exon per Megabase of library size) values were calculated using Cufflinks. Reads normalization, variance estimation and pair-wise differential expression analysis with multiple testing correction was conducted using the R Bioconductor DESeq2 package. Heatmaps and hierarchical clustering were generated with 'pheatmap' R package. GO enrichment analyses were performed using Genomatix. The RNA-seq data were deposited on GEO under the accession number GSE152416.

## Larval oral infection assay

Early third instar larvae were starved in empty vials for 2–3 hr, then placed in fly food vials containing banana pulp alone (uninfected control) or mixed with concentrated bacterial pellets (*E. coli* or *P. entemophila*) followed by incubation at 25°C. 10–12 hr post infection, larvae were washed in 70% ethanol, rinsed in water, dissected, and processed for immunostaining. *E. coli* (*Khadilkar et al., 2017a*); *P. entemophila* (provided by Sveta Chakrabarti, Indian Institute of Science).

## Wasp parasitism assay

Four to five female wasps (*Leptopilina boulardi,* provided by Tina Mukherjee, NCBS) were introduced into vials containing 40–50 late second instar (about 65 hr, AEL). Three to four hours post-infestation, wasps were removed from the vials and *Drosophila* larvae were further allowed to develop for 2, 3, or 4 days prior to dissections. Throughout the experiment larvae were maintained at 25°C, except for PSC ablation experiments (see below). Lamellocytes were counted manually on the basis of DAPI positive nuclei that showed lamellocyte-like morphology scored by Phalloidin or that expressed *MSNF9-mcherry*.

## Fly survival assay after wasp infestation

For survival assays, 40–50 late second instar larvae (about 65 hr AEL) were exposed to 4–5 female wasps (*Leptopilina boulardi*) at 25°C. Parasitized larvae identified by the presence of melanized spots were collected and allowed to develop at 25°C. The percentage of emerging adult flies was scored.

## PSC ablation

*Antp-Gal4* flies were crossed to *UAS-rpr,tub-GAL80^{ts}*, and their progenies were raised at 18°C until early second instar larval stage as early expression of Rpr driven by *Antp* is lethal at 25°C (*Benmimoun et al., 2015*). Larvae were then shifted to 29°C for 2 days and further allowed to develop at 25°C until dissection.

## Primers and in situ hybridization probes

For in situ hybridization, DIG-UTP labeled anti-sense RNA probes were used (*Avet-Rochex et al., 2010*). Required fragments were PCR-amplified from wild-type genomic DNA and used as template for preparation of DIG-labeled probe using T7 RNA Polymerase (Promega, USA) according to the manufacturer's instructions. Concentration of probe to be used for in situ hybridization assay was optimized with help of dot blot. Amplicon sizes and primers used for PCR amplification are: *latran*: (775 bp) forward primer 5′ CCACCCAGGGCAGCATGCTC 3′; reverse primer 5′ taatacgactcactatagggCCTATTGCGCTCATGGACAC 3′; *Tep1*: (769 bp) forward primer 5′ CCTTAGCCCTCAATCCGGCC 3′; *Tep1* reverse primer 5′ taatacgactcactatagggAACCGTCGTTACGTTTGTAG 3′. *Tep4* (802pb) forward primer 5′ CAGGGCAGAAGTTCAGAGGC 3′, reverse primer 5′ taatacgactcactatagggGTCCGCCAGCACCGGAATGG 3′. *upd3:* (550 bp) forward primer 5′ CCATTCCAGTTGAACCTTCG 3′, reverse primer 5′ taatacgactcactatagggATCGCCTTTGGCACGTGG 3′. *dome:* (791 bp) forward primer 5′ CTACGAGATCTCGCTGCGCG 3′, reverse primer 5′ taatacgactcactatagGGACGAGAAGGCCATGCCGC 3′. *Dystroglycan* (Dg) probe was generated using GH09323 cDNA (from DGRC) and SP6 RNA polymerase (Promega) for anti-sense transcription.

## Immunostaining, in situ hybridization, and microscopy

Staged larvae and pupae were used for dissection from timed embryos. Immunostainings were as described before (*Kulkarni et al., 2011*). Images were captured with Zeiss LSM 510 confocal or Zeiss LSM 880 confocal microscopes and analyzed using Zen black processing software and ImageJ. For in situ hybridization, lymph glands were fixed in 4% paraformaldehyde for 15 min, washed three times for 15 min each in PBST, and pre-incubated for 1 hr at 60°C in hybridization buffer (HB: 50% Formamide, 2× SSC, 1 mg/ml Torula RNA, 0.05 mg/ml Heparin, 2% Roche blocking reagent, 0.1% CHAPS, 5 mM EDTA, 0.1% Tween 20). Lymph gland preparations were then incubated overnight at 60°C with DIG-labeled RNA probe, followed by incubation for 1 hr in HB and for 30 min in 50% HB-50% PBST at 60°C and three washes 15 min each in PBST. 1% Goat serum was used as blocking agent for 30 min, followed by incubation with sheep anti-DIG antibody conjugated to alkaline phosphatase (Roche, Switzerland) (1:1000) for 2 hr. Larvae were extensively washed in PBST and the in

situ hybridization signal was revealed with NBT/BCIP substrate (Promega, USA). Alternatively, anti-DIG antibody coupled to horse-radish peroxidase (Roche, 1:1000) was used and the signal was revealed after extensive washing in PBST using TSA Plus Cyanine three system (Perkin Elmer). Lymph glands were mounted in 70% glycerol. Images were captured using Olympus IX70 bright field microscope or Leica LSM800 confocal microscope. To visualize the anterior and posterior lobes, two to three pictures (or Z-stacks) were captured along the A–P axis and stitched manually to reconstitute the whole lymph gland.

## Antibodies

Mouse anti-Antennapedia (1:20, DSHB #4C3), mouse anti-Hindsight (1:50, DSHB #1G9), rat anti-DE-cadherin (1:10, DSHB #DCAD2), mouse anti-Ubx (1:20; DSHB #FP3.38), mouse anti-Dlp (1:50, DSHB #13G8), mouse anti-Myospheroid (1:50, DSHB #CF.6G11), mouse anti-P1 antibody (1:100, kind gift from Istvan Ando, Biological Research Center of the Hungarian Academy of Sciences), mouse anti-collier antibody (1:100, kind gift from Michèle Crozatier, University of Toulouse), rabbit anti-GFP (1:1000, Clinisciences #TP401), chick or rabbit anti-GFP (1:500, Molecular Probes Inc), rabbit anti-DsRed (1:200, Takara Bio.), rabbit anti-V5 (1:1000, ThermoFisher #PA1-993), mouse anti-ProPO antibody (1:5, Bioneeds), and rabbit anti-Asrij (1:50, [*Kulkarni et al., 2011*]). Phalloidin was conjugated to Alexa-488 or Alexa-568 or Alexa-633 and secondary antibodies were Alexa-488, Alexa-568, or Alexa-633 conjugated (Molecular Probes, Inc).

## Image processing and analysis

For representative images, projections were made from complete Z-stacks and stitched manually to reconstitute the whole lymph gland. For Phalloidin, medial slices were stitched manually to avoid interference with the cardiac tube. Images were processed uniformly for brightness and contrast using Adobe Photoshop Elements 14. White dotted lines indicate lymph gland lobe boundaries, and yellow asterisks indicate pericardial cells. Complete Z-stacks were considered for all analysis. GFP mean fluorescence intensity (MFI) was measured using ImageJ software. The area to be measured for each lobe was marked with the help of the select tool, followed by intensity measurements. DAPI cell counts analysis was performed using 3D object counter module in ImageJ software. For cell cycle analysis 3D images were reconstructed for lymph gland lobes using complete Z-stacks with the help of IMARIS software. $GFP^+$, $RFP^+$ cells were analyzed using the spots module in IMARIS, and spots co-localization module was used to identify $GFP^+RFP^+$. For analysis of markers using 3D images, DAPI positive nuclei were counted by using the spots module in IMARIS. For membrane and cytoplasmic markers, surface module was used for 3D rendering. Distance between the 3D surface and spot was defined, with the help of modules – Find spots closer to surface and Find spots away from surface; number of nuclei that were positive or negative for a particular marker were determined. Quantifications were performed for the primary, secondary, and tertiary lobes individually.

## Statistical analysis and quantification

In all assays, control and test genotypes were analyzed in parallel. Each experiment was repeated independently at least three times. Graphs and statistical analyses were done by GraphPad Prism 8.0. Statistically significant differences were indicated by *$p < 0.05$, **$p < 0.01$, and ***$p < 0.001$ and ns indicates nonsignificant. Error bars represent standard error of mean (SEM).

## Acknowledgements

We thank the *Drosophila* Hemocyte Biology and European *Drosophila* Research Conference (EDRC) meeting participants for valuable discussions and feedback; *Drosophila* community for fly stocks and antibodies; National Centre for Biological Sciences, Fly Facility for stocks; JNCASR Imaging facility and our laboratory members for valuable inputs and suggestions. This work was funded by the Indo-French Centre for the Promotion of Advanced Research (IFCPAR/*CEFIPRA*) grant to MI and LW; MI's work was also supported by SERB grant, J C Bose award project and Jawaharlal Nehru Centre for Advanced Scientific Research. LW was supported by grants from the Agence Nationale pour la Recherche and Fondation ARC.

## Additional information

### Competing interests
K VijayRaghavan: Senior editor, *eLife*. The other authors declare that no competing interests exist.

### Funding

| Funder | Author |
|---|---|
| Indo-French Centre for the Promotion of Advanced Research | Lucas Waltzer<br>Maneesha S Inamdar |
| Science and Engineering Research Board | Maneesha S Inamdar |
| J C Bose Fellowship | Maneesha S Inamdar |
| Jawaharlal Nehru Centre for Advanced Scientific Research | Maneesha S Inamdar |
| Agence Nationale de la Recherche | Lucas Waltzer |

The funders had no role in study design, data collection and interpretation, or the decision to submit the work for publication.

### Author contributions
Diana Rodrigues, Data curation, Formal analysis, Validation, Investigation, Visualization, Methodology, Writing - original draft, Writing - review and editing; Yoan Renaud, Data curation, Formal analysis, Methodology, Writing - review and editing; K VijayRaghavan, Resources, Supervision, Writing - review and editing; Lucas Waltzer, Maneesha S Inamdar, Conceptualization, Resources, Data curation, Formal analysis, Supervision, Funding acquisition, Validation, Investigation, Visualization, Methodology, Writing - original draft, Project administration, Writing - review and editing

### Author ORCIDs
Diana Rodrigues ⬛ https://orcid.org/0000-0003-2395-4328
Yoan Renaud ⬛ http://orcid.org/0000-0002-4036-8315
K VijayRaghavan ⬛ http://orcid.org/0000-0002-4705-5629
Lucas Waltzer ⬛ https://orcid.org/0000-0002-5361-727X
Maneesha S Inamdar ⬛ https://orcid.org/0000-0002-8243-2821

### Decision letter and Author response
Decision letter https://doi.org/10.7554/eLife.61409.sa1
Author response https://doi.org/10.7554/eLife.61409.sa2

## Additional files

### Supplementary files
• Supplementary file 1. List of genes expressed in the lymph gland anterior and/or posterior lobes. The expression level of each gene (RPKM values) in each of the six RNA-seq samples (three anterior lobes and three posterior lobes) is indicated. Only genes with a FPKM $\geq 1$ in all three samples of the anterior lobes or of the posterior lobes are considered.

• Supplementary file 2. List of differentially expressed genes between posterior and anterior lobes (p<0.01, fold change >1.5). Blue: genes overexpressed in the posterior lobes. Red: genes overexpressed in the anterior lobes.

• Supplementary file 3. Over-represented GO categories (Biological Processes, Cellular Components, Molecular functions) among genes overexpressed in the anterior or posterior lobes.

• Transparent reporting form

## Data availability

RNA-seq data has been deposited in GEO under the accession number GSE152416. All data generated or analysed during this study are included in the manuscript and supporting files. Source data files have been provided for Figures 1, 3, 4, 5, 6, 7 and Figure 1—figure supplement 1, 2, Figure 4—figure supplement 1, 2, Figure 5—figure supplement 1, 2, Figure 7—figure supplement 1.

The following dataset was generated:

| Author(s) | Year | Dataset title | Dataset URL | Database and Identifier |
|---|---|---|---|---|
| Waltzer L, Renaud Y | 2021 | Transcriptomic analysis of Drosophila larval lymph glands | https://www.ncbi.nlm.nih.gov/geo/query/acc.cgi?acc=GSE152416 | NCBI Gene Expression Omnibus, GSE152416 |

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
