## [Decision Letter]

**Acceptance summary:**

The *Drosophila* lymph gland, a larval hematopoietic organ, is composed of multiple lobes. However, previous studies have been largely focused on the most anteriorly positioned lobe, called the primary lobe (anterior lobe). In this manuscript, Rodrigues et al. first shed light on how the posteriorly localized lymph gland lobes are regulated, and biological characteristics that distinguish posterior lobes from the anterior lobe. Moreover, this study also provides valuable genetic tools useful for analyzing the posterior lobes, laying the foundation for future studies.

**Decision letter after peer review:**

Thank you for submitting your article "Differential activation of JAK-STAT signaling reveals functional compartmentalization in *Drosophila* blood progenitors" for consideration by *eLife*. Your article has been reviewed by three peer reviewers, including Jiwon Shim as the Reviewing Editor and Reviewer #1, and the evaluation has been overseen by Utpal Banerjee as the Senior Editor.

The reviewers have discussed the reviews with one another and the Reviewing Editor has drafted this decision to help you prepare a revised submission.

Summary:

The reviewers found that the manuscript is nice and interesting and contains valid data. Given its future significance to the related field, the reviewers suggested to further strengthen the manuscript by supplementing detailed description/analyses of posterior lobes and investigating a more expanded variety of signaling pathways that might function in posterior lobes.

Essential revisions:

1) Separate the secondary and tertiary lobe: cell count, cell cycle, and phenotypes etc.

The authors start the paper with the fact that the posterior lobes contain a larger number of cells than the primary lobe and show that they maintain S to G2/M phase of the cell cycle similar to the primary lobe. To provide fundamental insights into the nature of posterior lobes, basic characteristics of the posterior lobes need to be analyzed with more details in addition to the expression analyses. For example, the number of secondary and tertiary lobes could be separately scored by which we can understand variations in the number of posterior lobes. The authors can also look into the developmental progression of their division (number and cell cycle) from early to late stages (cell counts of Figure 3 for pupal stages), which will explain how/when the posterior lobes acquire more cells as shown in Figure 1A. Moreover, FUCCI-based cell cycle patterns shown in the tertiary lobe could be different from the secondary lobe if separately counted (Figure 1H) and these will better represent the characteristics of the secondary/tertiary lobes.

In the same vein, more caution should be taken into describing expressions of markers and immune activated phenotypes in the secondary/tertiary lobes. Many gal4 expressions in this manuscript support the idea that the secondary lobe and the tertiary lobe may not be the same and cannot be lumped up as the posterior lobes/progenitors. For example, the tertiary lobe shows milder phenotypes than the secondary lobe when infested (Figures 4B, D, 5H-J, 6C-D, G-H), and that *domeMESO-GFP* is gone up primarily in the secondary lobe 4 days after infestation. These discrepancies are obvious in figures/graphs but are not clearly explained in the Results. Moreover, the secondary and tertiary lobes need to be separately quantitated in Figures 5A-f, K-M, 7E-f and K-L as in other graphs.

2) Genetic relationship between the PSC in AP and PP

While the authors focused a lot on how JAK-STAT is regulated intrinsically by different regulators in AP and PP, they seem not to consider that the biggest difference between AP and PP is not only their location but also the cell types they are connected with. AP is connected with hemocytes and the niche PSC, while PP is not at all. Hemocytes in cortical zone and the niche PSC all have been shown to actively secrete signals that control AP behavior. The authors should consider the contribution of hemocytes and PSC to the differential regulation of JAK-STAT in control and upon wasp using additional experiments. For example, does deleting the PSC delay/disrupt the differential regulation of JAK STAT.

3) Involvement of other pathways (Wg, Notch, NF-kB etc.)

The authors suggested that Upd3, Dlp, and Latran contribute to the differential regulation of JAK-STAT based on their findings (subsection “Differential regulation of the JAK-STAT inhibitor latran in lymph gland progenitors”): (1) Upd3 is expressed in both AP and PP, but upon wasp parasitism, only Upd3 expression in AP is reduced. (2) Dlp is expressed in PP but is low in AP (Figure 2D) and knockdown of Dlp enhances the response to parasitism in PP but not in AP. (3) Latran is expressed in both AP and PP. As seen in the images (Figure 7K-L), upon wasp parasitism latran expression is reduced in both AP and PP. The authors claim that these experiments provide potential mechanisms to explain why JAK-STAT is differentially regulated. These observations are also consistent with a model that there are differences upstream of JAK-STAT signaling that underlie the differences in response in the AP and PP to wasp infestation. While the author make a convincing argument that JAK-STAT signaling could account for some of these differences, it might not be the primary cause of differences in response. Consistent with this, multiple examples in the literature suggest Dlp is a general regulator of signaling. One way to address this is to see if the differences in Upd expression respond to changes to some of the many signaling pathways known to regulate hematopoiesis in the AP (such as Wg, Notch, hh etc.),

There are multiple pathways that mediate the immune response in progenitors upon wasp infestation. For example, Louradour et al., 2017, demonstrated that AP blood progenitors show an upregulated NF-kB transcriptional activity upon wasp infestation. However the authors only focused on JAK/STAT signaling. Is there an involvement of these other pathways? Answering this question is important because JAK/STAT might in fact act downstream of these other pathways. To address this question I would advise the authors look at least NF-kB pathway and monitor NF-kB transcriptional activity in both AP and PP in wild-type control and upon wasp parasitism. The authors should also examine possible cross-interaction of the two important immune pathways upon wasp parasitism using genetic interaction experiments.

4) The biological relevance of differential PP regulation

The authors revealed functional compartmentalization in the AP and PP of the lymph gland upon wasp parasitism. However, the actual contribution of this differential regulation to animal survival or systemic immune response upon immune challenges seems to be still unclear. Does the differential regulation of the JAK STAT pathway in AP and PP confer to post-survival advantage of flies upon immune challenge? Why is the induction of lamellocytes in PP inhibited even in the presence of wasp eggs? One may expect that the organism produces as many lamellocytes as possible to fight infestation. The authors should do simple experiments to test if induction of lamellocyte formation specifically in AP, PP, or both (for example using STAT RNAi) increases/decreases the survival rate of flies post-wasp parasitism and discuss this question in the Discussion section.

---

## [Author Response]

Essential revisions:1) Separate the secondary and tertiary lobe: cell count, cell cycle, and phenotypes etc.The authors start the paper with the fact that the posterior lobes contain a larger number of cells than the primary lobe and show that they maintain S to G2/M phase of the cell cycle similar to the primary lobe. To provide fundamental insights into the nature of posterior lobes, basic characteristics of the posterior lobes need to be analyzed with more details in addition to the expression analyses. For example, the number of secondary and tertiary lobes could be separately scored by which we can understand variations in the number of posterior lobes. The authors can also look into the developmental progression of their division (number and cell cycle) from early to late stages (cell counts of Figure 3 for pupal stages), which will explain how/when the posterior lobes acquire more cells as shown in Figure 1A. Moreover, FUCCI-based cell cycle patterns shown in the tertiary lobe could be different from the secondary lobe if separately counted (Figure 1H) and these will better represent the characteristics of the secondary/tertiary lobes.In the same vein, more caution should be taken into describing expressions of markers and immune activated phenotypes in the secondary/tertiary lobes. Many gal4 expressions in this manuscript support the idea that the secondary lobe and the tertiary lobe may not be the same and cannot be lumped up as the posterior lobes/progenitors. For example, the tertiary lobe shows milder phenotypes than the secondary lobe when infested (Figures 4B, D, 5H-J, 6C-Dd, G-H), and that domeMESO-GFP is gone up primarily in the secondary lobe 4 days after infestation. These discrepancies are obvious in figures/graphs but are not clearly explained in the Results. Moreover, the secondary and tertiary lobes need to be separately quantitated in Figures 5A-F, 5K-M, 7E-F, and K-L as in other graphs.

We appreciate the detailed comments and we have complied with those suggestions in the revised manuscript.

(1a) We have now scored the number of secondary and tertiary lobes across different development stages in the lymph gland to shed light on the variation in the number of posterior lobes and represented the detailed analysis in revised Figure 1—figure supplement 1A. Since very few larvae have quaternary lobes we have clubbed the analysis for the tertiary and the quaternary lobes for all cell counts.

(1b) As suggested by the reviewers, we also analyzed cell numbers separately for the primary, secondary and tertiary lobes at different times of development: 60h after egg laying (AEL) (mid second instar), 72h (early third instar), 96h (mid third instar), 120h (late third instar) and 144h (late wandering larvae). We observed that at 60h AEL secondary lobes consist of approximately 10-15 cells arranged in a thin row, tertiary lobes are hardly visible and quaternary lobes are absent (revised Figure 1A, Figure 1—figure supplement 1B-F). From 60h to 96h AEL we found that lymph gland secondary and tertiary lobes contain significantly fewer cells as compared to the primary lobes. By 120h AEL there is a strong increase in the number of cells in the posterior lobes and at 144h AEL posterior lobes together consist of about twice the number of cells as compared to primary lobes. These data bring novel insight into the development of the lymph gland lobes. The text has been edited appropriately to reflect this.

(1c) We also provide a thorough analysis of the cell cycle patterns of the progenitors across each lobe and at different time points. We used the FUCCI system in combination with *e33c-Gal4* (which drives expression in most cells of the lymph gland) as well as with *Tep4-Gal4* (driving expression in progenitors in the primary, secondary and, partially in tertiary/ quaternary lobes) to quantify the lymph gland cells and progenitors in G1, S or G2/M at 72h AEL, 96h AEL, 120h AEL and 144h AEL. These data are displayed in revised Figure 1H and revised Figure 1—figure supplement 2A-G. We found primary and posterior lobes were at different phases of the cell cycle in concordance with their developmental status. Analysis of *Tep4*-expressing progenitors, which form the major fraction of the progenitor pool, showed that most of the AP are in S phases from 72h to 120h AEL and that their proliferation is reduced at 144h AEL (revised Figure 1H and Figure 1—figure supplement 2A-C). In contrast, fewer cells are in S phase among the PP, and the proportion of PP in G1 phase constantly decreases from 72h to 144h AEL essentially to the benefit of cells in G2/M, both in the secondary and tertiary lobes. Cell cycle of the entire pool of lymph gland cells using *e33c-Gal4* driver showed that the overall distribution of proliferative cells was more in the tertiary lobes at 120h AEL (revised Figure 1—figure supplement 2D-G). These data bring further evidence of the heterogenous nature of the lymph gland progenitors as AP and PP exhibit distinct cell cycle pattern.

(1d) For better insight into the evolution of lymph gland during the pupal stages, we now also analyze the number of primary, secondary and tertiary lobes that have histolyzed, begin to histolyze, or are intact at 0, 5, 10 and 15h APF. We also provide a quantification of the proportion of cells that express the niche marker *collier*, the progenitor markers – *dome* and *Tep4*, or the differentiation markers- *Pxn* and P1/NimC1 in the primary, secondary and tertiary lobes, at 5h and 10h after pupal formation. These analyses confirm our previous conclusion that posterior progenitors are maintained up to at least 10h of pupal development (revised Figure 3).

2) Genetic relationship between the PSC in AP and PPWhile the authors focused a lot on how JAK-STAT is regulated intrinsically by different regulators in AP and PP, they seem not to consider that the biggest difference between AP and PP is not only their location but also the cell types they are connected with. AP is connected with hemocytes and the niche PSC, while PP is not at all. Hemocytes in cortical zone and the niche PSC all have been shown to actively secrete signals that control AP behavior. The authors should consider the contribution of hemocytes and PSC to the differential regulation of JAK-STAT in control and upon wasp using additional experiments. For example, does deleting the PSC delay/disrupt the differential regulation of JAK STAT.

Thank you for raising this important point. It is true that anterior progenitors (AP) receive signals from the differentiated hemocytes residing in the cortical zone and from the PSC/ niche, while the PP are physically separated from these cells and may not be subject to their inputs. We now stress this point in the revised Discussion.

As the PSC is required for the (systemic) response to wasp parasitism and in particular for lamellocyte differentiation in the anterior lobes (e.g. Crozatier et al., 2004; Sinenko et al., 2011; Benmimoun et al., 2015), it may indeed contribute to the differential response between AP and PP. Consistent with previous findings, we find that ablating the PSC by overexpressing the pro-apoptotic gene *reaper* using *Antp-Gal4* prevents lamellocyte differentiation in the primary lobes and we did not observe lamellocyte induction in the posterior lobes either in these conditions. Interestingly, AP of PSC-less larvae did not behave as PP of wild type larvae as they did not exhibit cell coalescence, a frequent PP response to wasp parasitism. Although we could not introduce a JAK/STAT reporter in these experiments to assess JAK/STAT pathway levels/regulation (we could not obtain viable larvae carrying *Antp-Gal4*, 10xSTAT-GFP and *UAS-reaper*, even at 18°C), our analysis suggests that the PSC is not responsible for the differential response between AP and PP. In particular, it is not required to prevent lamellocyte induction in the PP. Intriguingly though, PSC ablation prevented PP cell coalescence, indicating that it could be involved in PP response to parasitism, may be through systemic signaling as described for the induction of lamellocytes in circulating blood cells (Sinenko et al., 2011). We have presented these data in revised Figure 4—figure supplement 2.

Concerning the hemocytes of cortical zone, while they contribute to AP homeostasis in normal conditions (e.g. Mondal et al., 2011; Zhang and Cadigan, 2017), it is not known whether they also regulate AP response to wasp parasitism, but this is an interesting possibility. However due to lack of specific Gal4 drivers we cannot distinguish between the cortical zone (CZ) and the circulating hemocytes on AP and PP response. Indeed available drivers such as Hml-Gal4 or *Pxn-Gal4* drive expression in CZ as well as circulating/ sessile hemocyte compartments. It is thus not possible to effectively address the role of the CZ (or circulating hemocytes) on AP/PP response to wasp parasitism. Nonetheless we have discussed a possible involvement of the CZ in the differential response of AP and PP in the revised Discussion.

3) Involvement of other pathways (Wg, Notch, NF-kB etc.)The authors suggested that Upd3, Dlp, and Latran contribute to the differential regulation of JAK-STAT based on their findings (subsection “Differential regulation of the JAK-STAT inhibitor latran in lymph gland progenitors”): (1) Upd3 is expressed in both AP and PP, but upon wasp parasitism, only Upd3 expression in AP is reduced. (2) Dlp is expressed in PP but is low in AP (Figure 2D) and knockdown of Dlp enhances the response to parasitism in PP but not in AP. (3) Latran is expressed in both AP and PP. As seen in the images (Figure 7K-L), upon wasp parasitism latran expression is reduced in both AP and PP. The authors claim that these experiments provide potential mechanisms to explain why JAK-STAT is differentially regulated. These observations are also consistent with a model that there are differences upstream of JAK-STAT signaling that underlie the differences in response in the AP and PP to wasp infestation. While the author make a convincing argument that JAK-STAT signaling could account for some of these differences, it might not be the primary cause of differences in response. Consistent with this, multiple examples in the literature suggest Dlp is a general regulator of signaling. One way to address this is to see if the differences in Upd expression respond to changes to some of the many signaling pathways known to regulate hematopoiesis in the AP (such as Wg, Notch, hh etc.),There are multiple pathways that mediate the immune response in progenitors upon wasp infestation. For example, Louradour et al., 2017, demonstrated that AP blood progenitors show an upregulated NF-kB transcriptional activity upon wasp infestation. However the authors only focused on JAK/STAT signaling. Is there an involvement of these other pathways? Answering this question is important because JAK/STAT might in fact act downstream of these other pathways. To address this question I would advise the authors look at least NF-kB pathway and monitor NF-kB transcriptional activity in both AP and PP in wild-type control and upon wasp parasitism. The authors should also examine possible cross-interaction of the two important immune pathways upon wasp parasitism using genetic interaction experiments.

We are glad that the reviewers are convinced that JAK-STAT signaling could account for some of the differences between lobes. We do not intend to imply that it is the primary cause. As this is the first indepth study of posterior progenitors and mechanisms regulating them, interrogating all possible pathways is out of the scope of this manuscript.

Nevertheless we agree that multiple signaling pathways may regulate immune response in the PP in response to wasp parasitism. Here we show that PP resists differentiation in response to immune stress as compared to AP. Since JAK-STAT signalling has been implicated in progenitor maintenance and mediating immune response, we focused on this pathway. Our analyses suggest Dlp promotes differential activation of JAK-STAT signaling leading to differences in immune response across AP and PP. While Dlp is a regulator of the JAK-STAT pathway, it has also been implicated in Hedgehog, Wingless and Decapentaplegic pathways (Baeg and Perrimon, 2001; Cadigan 2002; Yan and Lin, 2009). Yet, the role of Hedgehog, Wingless and Decapentaplegic in regulating AP immune response to wasp parasitism has not been studied previously hence, characterizing these pathways in the AP and PP will involve major experiments that are beyond the scope of this manuscript. However, keeping in mind the reviewer’s concerns we analyzed Toll/NF-kB and Notch signaling, since these pathways have been studied in the AP in the context of wasp parasitism.

(3a) It was shown that Toll/NF-kB activation occurs in the anterior lobes progenitors and in the PSC to promote lamellocyte fate upon wasp infestation (Louradour et al., 2017; Gueguen et al., 2013). Using *Drosomycin-GFP (Drs-GFP)* reporter line (Ferrandon et al., 1998) we observe activation of Toll/NF-kB signaling in response to wasp parasitism in the AP as reported (Louradour et al., 2017; Gueguen et al., 2013) but also in the PP, although PP show little lamellocyte differentiation. These results thus indicate that the differential response of the PP to wasp infestation between anterior and posterior lobes is unlikely to involve this pathway. We have presented this data in revised Figure 5A-B (*Drs-GFP*).

(3b) Wasp parasitism was found to inhibit Notch activity, resulting in fewer crystals cells and increased lamellocyte differentiation in the AP (Small et al., 2014). Hence we also tested whether this pathway was differentially regulated in AP and PP. We used the Notch responsive element (*NRE-GFP*) line to assess Notch activity (Saj et al., 2010). In response to wasp infestation primary lobes show reduced Notch activity as reported (Small et al., 2014). However, PP barely express this reporter in control conditions, and upon infestation we observed a few *NRE-GFP-*expressing cells in the secondary lobes of some larvae but not in the tertiary lobes. This suggests that there is no major involvement of Notch pathway in PP upon wasp parasitism. Data are presented in revised Figure 5C-D.

4) The biological relevance of differential PP regulationThe authors revealed functional compartmentalization in the AP and PP of the lymph gland upon wasp parasitism. However, the actual contribution of this differential regulation to animal survival or systemic immune response upon immune challenges seems to be still unclear. Does the differential regulation of the JAK STAT pathway in AP and PP confer to post-survival advantage of flies upon immune challenge? Why is the induction of lamellocytes in PP inhibited even in the presence of wasp eggs? One may expect that the organism produces as many lamellocytes as possible to fight infestation. The authors should do simple experiments to test if induction of lamellocyte formation specifically in AP, PP, or both (for example using STAT RNAi) increases/decreases the survival rate of flies post-wasp parasitism and discuss this question in the Discussion section.

We agree it is important to understand the relevance of differential activation of JAK-STAT signaling and lamellocyte inhibition in the PP post-parasitism. In this context, we have already shown that *Stat92E* knockdown in AP and PP, driven by *dome-Gal4* or *Tep4-Gal4* leads to reduced survival in response to wasp infestation (Figure 5—figure supplement 1A, B; revised Figure 6—figure supplement 1A, B). Post-parasitism, AP downregulate JAK-STAT signaling and differentiate to lamellocytes (Makki et al., 2010). Therefore our results suggest that preventing lamellocyte differentiation of PP is essential for survival, which could reflect a function for these undifferentiated PP in fighting the parasite (for example by producing the complement-like factor Tep1) or at later stage for proper development (for instance at metamorphosis).

To substantiate our results, we perturbed JAK-STAT signaling in the AP and/or PP using different Gal4 drivers expressed widely in the lymph gland (*e33c-Gal4*; Harrison et al., 2005, Kulkarni et al., 2011) or in (part of) the posterior lobes (*dlp-Gal4* and *UbxM3-Gal4,* as identified in this work, cf. Revised Figure 2M, O) and we scored the proportion of larvae that survive to adulthood. We found that knocking down *dlp* using *e33c-Gal4,* decreased survival post-parasitism as compared to parasitized Gal4 controls (Figure 7M). Moreover, attenuating JAK-STAT signaling by knocking down *stat92E* in the PP using *dlp-Gal4* or *Ubx(M3)-Gal4* also leads to reduced survival (Figure 7N-O). These data, in line with our previous results (Figure 5—figure supplement 1A, B; revised Figure 6—figure supplement 1A, B), strongly suggest that increased activation of JAK-STAT signaling in the PP or PP subsets improves adult survival rate in response to wasp parasitism. They also underline the biological relevance of the PP differentiation regulation. We have presented this data in revised Figure 7M-O and incorporated it in the text appropriately.